# On Testing Conditional Mean Independence for Manifold-Valued Data

**Meiling Zeng** [1]   **Jinhong You** [1]   **Jicai Liu** [2]   **Shouxia Wang** [1]

## Abstract

This paper introduces a nonparametric test for conditional mean independence between a manifold-valued $Y$ and Euclidean predictors $X$. The test is built on a new measure called the Manifold Martingale Difference Divergence (MMDD), which characterizes conditional mean dependence by projecting observations onto the tangent space via the logarithmic map. We provide an empirical estimator for the MMDD, establish its asymptotic null distribution, and implement a wild bootstrap procedure for finite-sample inference. Simulations on three representative manifolds demonstrate that the proposed test maintains correct size under the null even when the distribution of $Y$ depends on $X$, in contrast to the severe size distortion exhibited by the distance covariance (dCov) test. At the same time, it achieves competitive power across a range of alternatives. An application to real data illustrates its practical utility.

## 1. Introduction

Modern data acquisition in fields such as neuroscience and medical imaging increasingly yields observations that are inherently non-Euclidean. Random objects may take the form of symmetric positive definite matrices (SPD), directional data on spheres, or elements of the Wasserstein space of distributions. Such data naturally reside on manifolds, where the standard Euclidean vector space structure—and the statistical methodology built upon it—does not directly apply. A fundamental challenge is to model and infer the relationship between these manifold-valued responses $Y$ and Euclidean covariates $X$, e.g., age, genetic markers, or treatment doses.

[1]School of Statistics and Data Science, Shanghai University of Finance and Economics, Shanghai, China [2]School of Statistics and Mathematics, Shanghai Lixin University of Accounting and Finance, Shanghai, China. Correspondence to: Jicai Liu <liujicai1234@126.com>, Shouxia Wang <wangshouxia@sufe.edu.cn>.

*Proceedings of the 43$^{rd}$ International Conference on Machine Learning*, Seoul, South Korea. PMLR 306, 2026. Copyright 2026 by the author(s).

For responses on a finite-dimensional Riemannian manifold, the local Euclidean structure of its tangent spaces permits a direct extension of classical regression ideas. Regression models for this setting have been well developed (Chang, 1989; Fisher, 1995; Prentice, 1989). Subsequent intrinsic models include parametric geodesic regression (Cornea et al., 2017; Thomas Fletcher, 2013), semiparametric regression (Shi et al., 2009) and nonparametric local kernel regression (Davis et al., 2010; Hinkle et al., 2012; Pelletier, 2006). Alternatively, the extrinsic regression framework has been formalized by Lin et al. (2017), building on the concept of extrinsic means (Patrangenaru & Ellingson, 2016). A more general framework, Fréchet regression (Petersen & Müller, 2019), extends the paradigm to responses in arbitrary metric spaces. To address the curse of dimensionality in high-dimensional settings, recent advances include the Fréchet single-index model and its associated asymptotic inference theory (Ghosal et al., 2023; Bhattacharjee & Müller, 2023). Collectively, these studies have established a comprehensive framework for modeling the conditional mean when the response $Y$ lies on a manifold.

An essential yet understudied problem within this modeling paradigm is to assess whether the covariate $X$ indeed contributes to the conditional mean of $Y$. This foundational question is formalized as conditional mean independence testing, which evaluates the null hypothesis

$$H_0 : \mathbb{E}(Y|X) = \mathbb{E}(Y) \quad \text{a.s.} \tag{1}$$

Existing work on testing dependencies for manifold-valued random objects has concentrated predominantly on assessing global independence. The seminal concept of distance covariance (dCov), introduced by Székely et al. (2007) for Euclidean random vectors, provides a foundational measure of dependence. Lyons (2013) extended this framework to general metric spaces, thereby facilitating global independence tests for data residing on manifolds. Leveraging this extension, Yan et al. (2025) developed a distance covariance-based screening procedure for high-dimensional settings within metric spaces. However, these methods target global dependence and cannot isolate conditional mean dependence—the primary quantity of interest in regression analysis for manifold-valued responses.

In parallel, a separate line of work addresses conditional mean independence testing in linear spaces. For Euclidean

data, Shao & Zhang (2014) introduced the martingale difference divergence (MDD) and its standardized version (MDC), which were adopted for high-dimensional variable screening (MDC-SIS). Su & Zheng (2017) employed MDD for regression specification testing. To address functional data, Lee et al. (2020) generalized this approach to Hilbert spaces via functional MDD (FMDD). Yet MDD-type methods rely on the linear structure of Euclidean or Hilbert spaces and are not directly applicable to manifold.

To date, no method exists to test conditional mean independence for manifold-valued responses. Classical dCov only measures global dependence, while MDD/FMDD cannot be directly adapted to manifold-valued responses. This critical gap motivates our work.

In this paper, we propose a novel nonparametric testing procedure to evaluate the null hypothesis $H_0$, where the response $Y$ is manifold-valued and the covariates $X$ take values in a Euclidean vector space. The main contributions of this paper are summarized as follows: (1) We generalize the martingale difference divergence to manifold-valued responses by integrating the logarithmic map and tangent space projection, providing a theoretically justified measure of conditional mean dependence for non-Euclidean data. (2) We derive an empirical estimator of the proposed MMDD statistic and characterize its asymptotic distribution under the null hypothesis and local alternatives (attaining non-trivial power for local alternatives converging to the null at rate $n^{-1/2}$). (3) We develop a computationally efficient wild bootstrap method to approximate the critical value, proving its consistency and validating its finite-sample performance. (4) Through extensive numerical simulations and a real data application to wind direction (spherical response) analysis, we demonstrate that MMDD outperforms dCov in identifying regression-relevant conditional mean dependence and exhibits robust size and power across diverse manifold types.

The rest of this paper is organized as follows. We introduce necessary geometric preliminaries in Section 2.1 and formally define the proposed Manifold Martingale Difference Divergence (MMDD) in Section 2.2. We study the asymptotic properties of the MMDD test statistic in Section 2.3. In Section 3, we develop the wild bootstrap procedure to approximate the test critical values. Section 4 presents extensive numerical simulations to compare the finite-sample performance of our proposed test with existing methods. In Section 5, we apply the proposed method to a real wind direction dataset. Section 6 concludes with a brief discussion. All technical proofs are provided in the Appendix A.

## 2. Manifold Martingale Difference Divergence

### 2.1. Preliminaries

Let $(\mathcal{M}, d)$ be a manifold space, where $d$ denotes the metric on $\mathcal{M}$. We consider a random pair $(X, Y)$ with covariates $X \in \mathbb{R}^p$ and response $Y \in \mathcal{M}$. In this general setting, $Y$ is referred to as a random object. Fréchet (1948) extended the conventional notions of mean and conditional mean to random objects in metric spaces, where

$$\mu := \mathbb{E}(Y) = \arg \min_{\omega \in \mathcal{M}} \mathbb{E}\left(d^2(Y, \omega)\right), \qquad (2)$$

$$m_X := \mathbb{E}(Y|X) = \arg \min_{\omega \in \mathcal{M}} \mathbb{E}\left(d^2(Y, \omega)|X\right) \qquad (3)$$

are defined as the Fréchet mean of $Y$ and the conditional Fréchet mean of $Y$ given $X$, respectively.

At the point $\mu$, the tangent space $T_\mu \mathcal{M}$ of the manifold $\mathcal{M}$ is a Euclidean space, which represents a first-order approximation of $\mathcal{M}$ near $\mu$. The projection of $Y$ onto $T_\mu \mathcal{M}$, denoted by $\mathrm{Log}_\mu(Y)$, can be interpreted as the difference between $Y$ and $\mu$, and is also referred to as the logarithmic map on the manifold. When $\mathcal{M}$ is a Euclidean space, this projection reduces to the standard vector subtraction $\mathrm{Log}_\mu(Y) = Y - \mu$. The null hypothesis (1) is equivalent to

$$H_0 : \mathbb{E}\left(\mathrm{Log}_\mu(Y)|X\right) = 0 \quad \text{a.s.} \qquad (4)$$

The equivalence between (1) and (4) is shown in Appendix A.1. Correspondingly, the alternative hypothesis is

$$H_1 : \Pr\{\mathbb{E}\left(\mathrm{Log}_\mu(Y)|X\right) = 0\} < 1. \qquad (5)$$

### 2.2. Test Statistic

To introduce our new metric Manifold Martingale Difference Divergence (MMDD) for manifold-valued data, we briefly review FMDD. For random elements $Y \in \mathcal{L}_Y$ and $X \in \mathcal{L}_X$, where $\mathcal{L}_Y$ and $\mathcal{L}_X$ are separable Hilbert spaces, Lee et al. (2020) proposed the Functional Martingale Difference Divergence (FMDD) to quantify the contribution of covariate $X$ to the conditional mean of $Y$. The FMDD is defined as

$$\mathrm{FMDD}(Y|X) = -\mathbb{E}(\langle Y - \mu_Y, Y' - \mu_Y \rangle \|X - X'\|),$$

where $\mu_Y$ is the mean function of $Y$ and $(X', Y')$ is an i.i.d. copy of $(X, Y)$.

However, FMDD relies on linear operations (subtraction, inner product) inherent to Hilbert spaces. This motivates our MMDD, which adapts the paradigm to manifold data via tangent spaces. Let $K : \mathbb{R}^p \to \mathbb{R}$ be a positive semi-definite kernel. For clarity, this paper focuses on translation-invariant kernel functions, including the Gaussian and Laplace kernels.

**Definition 2.1** (Manifold Martingale Difference Divergence). For $Y \in \mathcal{M}$ and $X \in \mathbb{R}^p$, we define

$$
\begin{aligned}
\mathrm{MMDD}(Y|X) = \mathbb{E}\Big( &\langle \mathrm{Log}_\mu(Y), \mathrm{Log}_\mu(Y') \rangle_{T_\mu \mathcal{M}} \\
&\times K(X - X') \Big),
\end{aligned} \tag{6}
$$

where $\mu$ is the Fréchet mean of $Y$, $\langle \cdot, \cdot \rangle_{T_\mu \mathcal{M}}$ denotes the inner product on the tangent space $T_\mu \mathcal{M}$ and $(X', Y')$ is an independent copy of $(X, Y)$.

*Remark* 2.2. MMDD extends the intrinsic covariance on manifolds $\mathbb{E}(\langle \mathrm{Log}_\mu(Y), \mathrm{Log}_\mu(Y) \rangle_{T_\mu \mathcal{M}})$ (Pennec, 2006) to a kernel-weighted conditional setting, following the covariance-based construction of MDD (Euclidean space) and FMDD (Hilbert spaces), where analogous terms correspond to the covariance of $Y$ in their respective spaces.

The following proposition establishes key properties of MMDD, which underpin our conditional mean independence test.

**Proposition 2.3.** *For $Y \in \mathcal{M}$ and $X \in \mathbb{R}^p$, suppose that $\mathbb{E}(\|\mathrm{Log}_\mu(Y)\|_{T_\mu \mathcal{M}}) < \infty$. Then, we have*

1. $\mathrm{MMDD}(Y|X) \geq 0$;

2. $\mathrm{MMDD}(Y|X) = 0$ *if and only if $H_0$ is true.*

Proposition 2.3 implies that testing $H_0$ is equivalent to testing whether $\mathrm{MMDD}(Y|X) = 0$. In practice, the Fréchet mean $\mu$ of $Y$ is unknown. Given the independent and identically distributed (i.i.d.) observations $\{(X_i, Y_i)\}_{i=1}^n$, we estimate $\mu$ using the empirical Fréchet mean

$$
\hat{\mu} = \arg \min_{\omega \in \mathcal{M}} \frac{1}{n} \sum_{i=1}^n d^2(Y_i, \omega). \tag{7}
$$

A natural empirical estimator of $\mathrm{MMDD}(Y|X)$ is defined as

$$
\begin{aligned}
&\mathrm{MMDD}_n(Y|X) \\
&= \frac{1}{n(n-1)} \sum_{i \neq j} \langle \mathrm{Log}_{\hat{\mu}}(Y_i), \mathrm{Log}_{\hat{\mu}}(Y_j) \rangle_{T_{\hat{\mu}} \mathcal{M}} k_{i,j}, 
\end{aligned} \tag{8}
$$

where $k_{i,j} = K(X_i - X_j)$.

## 2.3. Asymptotic Properties

**Assumption 2.4.** The manifold $\mathcal{M}$ and the manifold-valued response $Y$ satisfy either of the following geometric regularity conditions:

(M1) $\mathcal{M}$ is a simply connected and complete manifold, with bounded non-positive sectional curvatures.

(M2) $\mathcal{M}$ is a simply connected and complete subset of a complete Riemannian manifold with positive sectional curvatures upper bounded by $\kappa > 0$, and satisfies a bounded diameter condition: $\sup_{p,q \in \mathcal{M}} d_{\mathcal{M}}(p, q) < \pi / \kappa^{1/2}$.

**Assumption 2.5.** The empirical Fréchet mean $\hat{\mu}$ satisfies the first-order asymptotic expansion

$$
\mathrm{Log}_\mu(\hat{\mu}) = A^{-1} \frac{1}{n} \sum_{k=1}^n \mathrm{Log}_\mu(Y_k) + o_p(n^{-1/2}),
$$

where the linear operator $A : T_\mu \mathcal{M} \to T_\mu \mathcal{M}$ is defined as the Hessian of the population Fréchet functional $F(\omega) = \mathbb{E}[d^2(Y, \omega)]$ at its minimizer $\omega = \mu$. Equivalently, $A = \nabla^2 F(\mu)$, which is a positive-definite operator on the tangent space $T_\mu \mathcal{M}$.

Assumption 2.4 guarantees the uniqueness of the Fréchet mean $\mu$ in Eq.(2) and ensures the logarithmic map $\mathrm{Log}_\mu(Y)$ is well-defined. Assumption 2.5, in turn, ensures that the empirical Fréchet mean $\hat{\mu}$ in Eq.(7) is $\sqrt{n}$-consistent and asymptotically normal for $\mu$, as established in the classical work of Bhattacharya & Patrangenaru (2003; 2005). Building on these two assumptions, the following theorem establishes the asymptotic distribution of the sample estimator $\mathrm{MMDD}_n(Y|X)$.

**Theorem 2.6.** *Suppose that Assumptions 2.4 and 2.5 hold, and $\mathbb{E}(\|\mathrm{Log}_\mu(Y)\|_{T_\mu \mathcal{M}}^2) < \infty$. Then under $H_0$, we have*

$$
n \mathrm{MMDD}_n(Y|X) \xrightarrow{d} \sum_{\nu=1}^\infty \lambda_\nu G_\nu^2,
$$

*where $G_\nu \overset{i.i.d.}{\sim} \mathcal{N}(0, 1)$, and $\lambda_\nu$ are eigenvalues of the integral equation*

$$
\int_{-\infty}^\infty h(Z_1, Z_2) f_\nu(Z_2) dF(Z_2) = \lambda_\nu f_\nu(Z_1).
$$

*Here, $h(Z_1, Z_2)$ is defined in Eq.(11) in Appendix A.3, with $f_\nu(\cdot)$ and $F(\cdot)$ being the probability density function and the limiting cumulative distribution function of $Z_i = (X_i, \mathrm{Log}_\mu(Y_i))$.*

According to Theorem 2.6, we define our test statistic as

$$
T_n = n \mathrm{MMDD}_n(Y|X).
$$

To understand the behavior of $T_n$ when the null hypothesis does not hold, we study the limiting distribution of $T_n$ under the following local alternative

$$
H_{1,n} : \mathrm{Log}_\mu(Y) = n^{-a} g(X) + \varepsilon, \tag{9}
$$

for some $a > 0$, where $g : \mathbb{R}^p \to T_\mu \mathcal{M}$ satisfies $\mathbb{E}[g(X)] = 0$ and $\mathbb{E}(\langle g(X), g(X') \rangle_{T_\mu \mathcal{M}} K(X - X')) > 0$, and $\varepsilon \in T_\mu \mathcal{M}$ satisfies $\mathbb{E}(\varepsilon \mid X) = 0$ a.s. together with $\Pr\{\langle g(X), \varepsilon \rangle_{T_\mu \mathcal{M}} \neq 0\} > 0$.

**Theorem 2.7.** *Suppose that Assumptions 2.4 and 2.5 hold, and $\mathbb{E}(\|g(X)\|^2_{T_\mu\mathcal{M}} + \|\varepsilon\|^2_{T_\mu\mathcal{M}}) < \infty$. Under the local alternative $H_{1,n}$,*

1. *if $0 < a < 1/2$, we have $T_n \overset{p}{\to} \infty$;*

2. *if $a = 1/2$, we have*

$$T_n \overset{d}{\to} c + \sum_{\nu=1}^{\infty} \lambda_\nu G_\nu^2,$$

*where $c = \mathbb{E}[\langle g(X), g(X')\rangle_{T_\mu\mathcal{M}} K(X - X')] > 0$, $G_\nu$ and $\lambda_\nu$ are defined in Theorem 2.6 with $Z_i = (X_i, \varepsilon_i)$;*

3. *if $a > 1/2$, we have*

$$T_n \overset{d}{\to} \sum_{\nu=1}^{\infty} \lambda_\nu G_\nu^2.$$

Because $c > 0$, the limiting distribution under the alternative $c + \sum_{\nu=1}^{\infty} \lambda_\nu G_\nu^2$ (with $a = 1/2$) stochastically dominates the limiting null distribution $\sum_{\nu=1}^{\infty} \lambda_\nu G_\nu^2$. This implies our test has nontrivial asymptotic local power against local alternatives converging to the null hypothesis at rate $n^{-1/2}$. For a similar discussion, see Fan (1998).

# 3. Wild Bootstrap Test

Since the limiting null distribution of the proposed test statistic $T_n$ is nonpivotal, we propose a wild bootstrap procedure to approximate the null distribution and show its asymptotic validity. The detailed steps are as follows:

1. **Generate bootstrap statistics**: For the $b$-th bootstrap replication ($b = 1, 2, \dots, B$), compute

$\text{MMDD}_n^*(Y|X)^b$
$$= \frac{1}{n(n-1)} \sum_{i \neq j} \eta_i^{(b)} \langle \text{Log}_{\hat{\mu}}(Y_i), \text{Log}_{\hat{\mu}}(Y_j)\rangle_{T_{\hat{\mu}}\mathcal{M}} k_{i,j} \eta_j^{(b)},$$

where $\{\eta_i^{(b)}\}_{i=1}^n$ are i.i.d. random variables with $\mathbb{E}(\eta_i^{(b)}) = 0$ and $\text{Var}(\eta_i^{(b)}) = 1$, e.g., standard normal random variables. The corresponding bootstrap test statistic is defined as

$$T_{n,b}^* = n\text{MMDD}_n^*(Y|X)^b.$$

2. **Collect bootstrap replicates**: Repeat the above step for $B$ times to obtain the set of bootstrap statistics $\{T_{n,b}^*\}_{b=1}^B$.

3. **Determine critical value**: Compute the $(1 - \alpha)$-th quantile of $\{T_{n,b}^*\}_{b=1}^B$, denoted as $Q_{(1-\alpha),n}^*$, which serves as the critical value for the test at the significance level $\alpha$.

4. **Test decision**: Reject the null hypothesis $H_0$ if $T_n > Q_{(1-\alpha),n}^*$; otherwise, fail to reject $H_0$.

To analyze the asymptotic behavior of the bootstrap test statistic, we introduce notions of bootstrap orders and bootstrap convergence (adapted from Remark 1 in Chang & Park (2003) and Definition 2 in Li et al. (2003) ).

**Definition 3.1** (Bootstrap Order). Let $T_n^*$ be a bootstrap statistic dependent on the random sample $\{Z_i\}_{i=1}^n$. Define

1. $T_n^* = o_p^*(1)$ a.s. if for any $\epsilon > 0$, $\text{Pr}^*\{|T_n^*| > \epsilon\} \to 0$ a.s., where $\text{Pr}^*$ denotes the conditional probability given $\{Z_i\}_{i=1}^n$.

2. $T_n^* = O_p^*(1)$ a.s. if for any $\epsilon > 0$, there exists a constant $M > 0$ such that for large $n$, $\text{Pr}^*\{|T_n^*| > M\} < \epsilon$ a.s.

Notice that $o_p^*(1)$ and $O_p^*(1)$ are for bootstrap sample asymptotics, which have similar definitions to $o_p(1)$ and $O_p(1)$. We extend these to $O_p^*(c_n)$ and $o_p^*(c_n)$ for deterministic sequences $c_n$, analogously to classical $O_p(c_n)$ and $o_p(c_n)$.

**Definition 3.2** (Bootstrap Consistency). Let $T_n^*$ be a bootstrap statistic that depends on the random sample $(Z_i)_{i=1}^n$. We say $(T_n^* \mid Z_1, Z_2, \dots)$ converges to $(T \mid Z_1, Z_2, \dots)$ in distribution *almost surely* if $(T_n^* \mid Z_1, Z_2, \dots)$ converges in distribution to $(T \mid Z_1, Z_2, \dots)$ for almost every sequence $(Z_1, Z_2, \dots)$. The notation for this *almost sure convergence in distribution* is:

$$T_n^* \overset{d^*}{\to} T \text{ a.s.}$$

We now examine the asymptotic distribution of our bootstrap statistic $T_n^*$ under the null and local alternatives.

**Theorem 3.3.** *Suppose that Assumptions 2.4 and 2.5 hold, $\mathbb{E}(\|\text{Log}_\mu(Y)\|^4_{T_\mu\mathcal{M}}) < \infty$ and $\mathbb{E}(\eta^4) < \infty$. Then under $H_0$, we have*

$$T_n^* \overset{d^*}{\to} \sum_{\nu=1}^{\infty} \lambda_v G_\nu^2 \quad a.s.,$$

*where $\{\lambda_\nu, G_\nu\}_{\nu=1}^\infty$ are defined in Theorem 2.6.*

Hence, the wild bootstrap consistently approximates the limiting null distribution of the test statistic. To examine the power of the bootstrap-based test, we further characterize the asymptotic behavior of the bootstrap statistic under alternative hypotheses as follows.

**Theorem 3.4.** *Suppose that Assumptions 2.4 and 2.5 hold, $\mathbb{E}(\|g(X)\|^4_{T_\mu\mathcal{M}} + \|\varepsilon\|^4_{T_\mu\mathcal{M}}) < \infty$ and $\mathbb{E}(\eta^4) < \infty$. Then under the local alternative $H_{1,n}$,*

1. *if $0 < a < 1/2$, we have*

$$\text{Pr}\{T_n \geq Q_{(1-\alpha),n}^* \mid H_{1,n}\} \to 1,$$

*where $Q_{(1-\alpha),n}^*$ is the $(1 - \alpha)$-th quantile of the bootstrap test statistic.*

2. *if $a = 1/2$, we have*

$$\Pr\{T_n \geq Q^*_{(1-\alpha),n} \mid H_{1,n}\} \to \Pr\{\mathcal{G}_1 \geq Q_{(1-\alpha),\mathcal{G}_0} - c\},$$

*where $\mathcal{G}_1 = \sum_{\nu=1}^{\infty} \lambda_\nu G_\nu^2$ follows the asymptotic distribution of $T_n - c$ under $H_{1,n}$ when $a = 1/2$, and $Q_{(1-\alpha),\mathcal{G}_0}$ is the $(1-\alpha)$-th quantile of the limiting null distribution.*

3. *if $a > 1/2$, we have*

$$\Pr\{T_n \geq Q^*_{(1-\alpha),n} \mid H_{1,n}\} \to \alpha.$$

Theorem 3.4 underpins our wild bootstrap procedure and the simulation studies in Section 4. It shows that the test has non-trivial asymptotic power against $n^{-1/2}$-local alternatives, while the limiting null distribution is non-pivotal. This justifies the use of the bootstrap-based testing framework in practice.

# 4. Numerical Simulation

In this section, we study the finite-sample performance of MMDD for manifold data. The test statistic is constructed using a Gaussian kernel $K(z) = \exp(-\|z\|^2/(2h^2))$ with bandwidth $h = 0.5$. We compare our method with the dCov-based test (Székely et al., 2007), which relies on computing the dCov statistic followed by permutation testing with $n_{\text{perm}} = 499$ permutations. In our simulations, we set the nominal level $\alpha$ at 0.1, 0.05 and 0.01. For each example, the bootstrap sample size is set to $B = 499$, and the perturbation variables $\{\eta_i\}_{i=1}^n$ are generated as independent standard normal random variables. To compute the empirical size and power of the test, 1000 Monte Carlo replicates are generated for each scenario.

## 4.1. Data generating processes

We consider the following data generating processes (DGPs) built on a unified tangent space regression framework for all manifolds

$$\text{Log}_\mu(Y_i) = g(X_i) + h(X_i)\varepsilon_i,$$

where $g : \mathbb{R}^p \to T_\mu \mathcal{M}$ denotes the signal function that satisfies $\mathbb{E}[g(X)] = 0$, while $h : \mathbb{R}^p \to T_\mu \mathcal{M}$ represents the heteroscedasticity function that scales the noise. We specify $p$-dimensional covariates $X_i \sim \mathcal{N}(0, I_p)$ with $p = 5$, that is $X_i = (X_{i1}, X_{i2}, X_{i3}, X_{i4}, X_{i5})^\top$. For the random noise $\varepsilon_i$, we generate samples from $\mathcal{N}(0, \sigma^2)$ with a standard deviation of $\sigma = 0.4$. Specifically, we design the following five DGPs to evaluate the MMDD test.

**DGP1:** $g(X_i) = 0, h(X_i) = \exp\left(s^2(X_{i4} + X_{i5})/2\right)v_2$;

**DGP2:** $g(X_i) = r^2(\beta^\top X_i)v_1, \quad h(X_i) = v_2$;

**DGP3:** $g(X_i) = r^2(\beta^\top X_i)v_1,$
$h(X_i) = \exp\left(s^2(X_{i4} + X_{i5})/2\right)v_2$;

**DGP4:** $g(X_i) = r^2\left(X_{i1}^2 + \exp(X_{i2}) + \sin(X_{i3})\right)v_1,$
$h(X_i) = v_2$;

**DGP5:** $g(X_i) = r^2\left(X_{i1}^2 + \exp(X_{i2}) + \sin(X_{i3})\right)v_1,$
$h(X_i) = \exp\left(s^2(X_{i4} + X_{i5})/2\right)v_2,$

where $v_1$ and $v_2$ denote fixed tangent directions in $T_\mu \mathcal{M}$, $\beta = (1/\sqrt{p}, \ldots, 1/\sqrt{p})^\top$ is a unit coefficient vector, $r^2$ controls the signal strength of the signal function $g(X_i)$ and $s^2$ regulates the variance dependence of the heteroscedasticity function $h(X_i)$. Besides, we center $g(X_i)$ in the construction for all DGPs to ensure $\mathbb{E}[g(X)] = 0$.

DGP1 corresponds to the null hypothesis, maintaining conditional mean independence while allowing the variance of $Y$ to depend on $X$, in order to examine type I error control. DGP2 introduces a linear mean dependence with homoscedastic noise to evaluate test power against linear alternatives. To assess robustness under variance heterogeneity, DGP3 combines linear dependence with heteroscedastic noise. DGP4 incorporates a nonlinear dependence structure under homoscedastic noise in order to evaluate the statistical power for detecting complex nonlinear relationships. Finally, DGP5 combines a nonlinear dependence structure with heteroscedastic noise.

## 4.2. Simulation Results

We conduct simulation studies on three manifolds for the response $Y$, each equipped with its canonical metric: the sphere $\mathbb{S}^2$ (standard induced Riemannian metric); the Wasserstein space of Gaussian distributions $\mathcal{W}_2(\mathbb{R})$ (2-Wasserstein metric); and the space of $d \times d$ symmetric positive definite (SPD) matrices $\mathcal{P}_d$ (affine-invariant Riemannian metric, AIRM). These manifolds are widely studied in non-Euclidean data analysis and represent distinct geometric structures.

**Example 1: Sphere Manifold $\mathbb{S}^2$.** For the spherical manifold $\mathcal{M} = \mathbb{S}^2$, we set the Fréchet mean $\mu = (1, 0, 0)^\top$ and specify the fixed tangent vector $v_1 = (0, 1, 0)^\top$ and $v_2 = (0, 0, 1)^\top$ in $T_\mu \mathbb{S}^2$. For DGP1, we consider $s^2 = 0, 0.2, 0.4, 0.5, 0.7$ with a sample size $n = 100$, and compare the rejection rates of our MMDD-based test and the dCov-based test.

Table 1 presents the rejection rates of the two tests across different variance dependence levels. We observe that the MMDD-based test retains size control under conditional mean independence with variance dependence. Specifically, at $s^2 = 0.7$, its rejection rates $(0.112, 0.051, 0.008)$ remain close to the nominal levels $(0.1, 0.05, 0.01)$. In contrast, the dCov-based test exhibits severe size distor-

tion, with rejection rates sharply increasing with $s^2$ to $(0.987, 0.967, 0.871)$. This indicates its inability to separate mean from variance dependence, resulting in inflated type I errors under the null of conditional mean independence.

*Table 1.* Rejection rates of MMDD-based and dCov-based tests under DGP1 ($n = 100$) on $\mathbb{S}^2$.

| SIZE | | $\alpha = 0.1$ | $\alpha = 0.05$ | $\alpha = 0.01$ |
|---|---|---|---|---|
| $s^2 = 0$ | MMDD | 0.083 | 0.031 | 0.005 |
| $s^2 = 0.2$ | MMDD | 0.088 | 0.041 | 0.007 |
| $s^2 = 0.4$ | MMDD | 0.092 | 0.044 | 0.006 |
| $s^2 = 0.5$ | MMDD | 0.104 | 0.050 | 0.006 |
| $s^2 = 0.7$ | MMDD | 0.112 | 0.051 | 0.008 |
| $s^2 = 0$ | dCov | 0.096 | 0.053 | 0.008 |
| $s^2 = 0.2$ | dCov | 0.188 | 0.094 | 0.017 |
| $s^2 = 0.4$ | dCov | 0.618 | 0.476 | 0.172 |
| $s^2 = 0.5$ | dCov | 0.829 | 0.718 | 0.410 |
| $s^2 = 0.7$ | dCov | 0.987 | 0.967 | 0.871 |

*Table 2.* Rejection rates of MMDD-based test under DGP2, DGP3 ($s^2 = 0.5$), DGP4, DGP5 ($s^2 = 0.5$) on $\mathbb{S}^2$.

| POWER | $\alpha = 0.1$ | | $\alpha = 0.05$ | | $\alpha = 0.01$ | |
|---|---|---|---|---|---|---|
| | $n = 100$ | $n = 200$ | $n = 100$ | $n = 200$ | $n = 100$ | $n = 200$ |
| DGP2 | | | | | | |
| $r^2 = 0.1$ | 0.139 | 0.166 | 0.060 | 0.096 | 0.007 | 0.026 |
| $r^2 = 0.2$ | 0.279 | 0.545 | 0.172 | 0.386 | 0.029 | 0.134 |
| $r^2 = 0.3$ | 0.562 | 0.902 | 0.430 | 0.833 | 0.155 | 0.581 |
| $r^2 = 0.4$ | 0.825 | 0.997 | 0.713 | 0.988 | 0.376 | 0.930 |
| $r^2 = 0.5$ | 0.950 | 1.000 | 0.899 | 1.000 | 0.613 | 0.994 |
| DGP3 | | | | | | |
| $r^2 = 0.1$ | 0.133 | 0.163 | 0.058 | 0.087 | 0.006 | 0.024 |
| $r^2 = 0.2$ | 0.242 | 0.449 | 0.128 | 0.311 | 0.020 | 0.103 |
| $r^2 = 0.3$ | 0.528 | 0.796 | 0.355 | 0.683 | 0.092 | 0.396 |
| $r^2 = 0.4$ | 0.729 | 0.975 | 0.592 | 0.954 | 0.249 | 0.822 |
| $r^2 = 0.5$ | 0.896 | 0.998 | 0.808 | 0.995 | 0.511 | 0.961 |
| DGP4 | | | | | | |
| $r^2 = 0.05$ | 0.141 | 0.209 | 0.075 | 0.121 | 0.007 | 0.021 |
| $r^2 = 0.10$ | 0.338 | 0.632 | 0.213 | 0.499 | 0.029 | 0.226 |
| $r^2 = 0.15$ | 0.623 | 0.949 | 0.484 | 0.899 | 0.155 | 0.693 |
| $r^2 = 0.20$ | 0.839 | 0.994 | 0.743 | 0.991 | 0.376 | 0.943 |
| DGP5 | | | | | | |
| $r^2 = 0.05$ | 0.147 | 0.157 | 0.067 | 0.087 | 0.002 | 0.011 |
| $r^2 = 0.10$ | 0.291 | 0.584 | 0.171 | 0.401 | 0.032 | 0.166 |
| $r^2 = 0.15$ | 0.564 | 0.898 | 0.418 | 0.817 | 0.125 | 0.528 |
| $r^2 = 0.20$ | 0.797 | 0.983 | 0.669 | 0.963 | 0.315 | 0.854 |

For DGPs 2 and 4, and for DGPs 3 and 5 with variance-dependent noise set to $s^2 = 0.5$, we investigate the influence of the signal strength $r^2$ and the sample size $n$ on the test power. In the linear signal settings (DGPs 2 and 3), $r^2$ takes values $0.1, 0.2, 0.3, 0.4, 0.5$; for the nonlinear signal settings (DGPs 4 and 5), $r^2$ is chosen as $0.05, 0.1, 0.15, 0.2$. The sample sizes are $n = 100$ and $200$ throughout.

The results presented in Table 2 show that the empirical power of the MMDD-based test consistently increases with

stronger signals or larger sample sizes, under both linear and nonlinear signal designs. In the linear setting, when strong variance-dependent noise ($s^2 = 0.5$) is introduced in DGP3, its power is moderately lower than that for DGP2, yet without a substantial drop. This indicates the robustness of the MMDD-based test to variance dependency while still effectively capturing linear signal dependence. In the nonlinear setting, a similar pattern of robustness is observed for DGP5 compared to DGP4, with test power showing a consistent upward trend as signal strength or sample size increases, further confirming the effectiveness and stability of the MMDD-based test in detecting complex dependency.

**Example 2: Wasserstein Space of Gaussian Distributions** $\mathcal{W}_2(\mathbb{R})$. For the Wasserstein space of univariate Gaussian distributions $\mathcal{M} = \mathcal{W}_2(\mathbb{R})$, each element is a Gaussian density $\mathcal{N}(\cdot, \cdot)$. We set the Fréchet mean $\mu = \mathcal{N}(0, 1)$. Define fixed tangent vectors $v_1 = (1, 0)^\top$ and $v_2 = (0, 1)^\top$ in $T_\mu \mathcal{W}_2(\mathbb{R})$.

The results under DGP1 on $\mathcal{W}_2(\mathbb{R})$, as shown in Table 3, reaffirm the pattern observed on $\mathbb{S}^2$. The MMDD-based test successfully controls size under variance-dependent noise, while the dCov-based test exhibits substantial size distortion. The power performance under DGPs 2–5 on $\mathcal{W}_2(\mathbb{R})$ (see Table 4) exhibits the same consistent pattern observed on $\mathbb{S}^2$ in Example 1. The empirical power of the MMDD-based test increases monotonically with the signal strength $r^2$ and sample size $n$, and remains robust to the introduction of variance-dependent noise ($s^2 = 0.5$).

*Table 3.* Rejection rates of MMDD-based and dCov-based tests under DGP1 ($n = 100$) on $\mathcal{W}_2(\mathbb{R})$.

| SIZE | | $\alpha = 0.1$ | $\alpha = 0.05$ | $\alpha = 0.01$ |
|---|---|---|---|---|
| $s^2 = 0$ | MMDD | 0.089 | 0.043 | 0.005 |
| $s^2 = 0.2$ | MMDD | 0.090 | 0.041 | 0.006 |
| $s^2 = 0.4$ | MMDD | 0.096 | 0.040 | 0.006 |
| $s^2 = 0.5$ | MMDD | 0.100 | 0.042 | 0.008 |
| $s^2 = 0.7$ | MMDD | 0.110 | 0.044 | 0.008 |
| $s^2 = 0$ | dCov | 0.099 | 0.047 | 0.006 |
| $s^2 = 0.2$ | dCov | 0.175 | 0.091 | 0.013 |
| $s^2 = 0.4$ | dCov | 0.547 | 0.356 | 0.141 |
| $s^2 = 0.5$ | dCov | 0.732 | 0.564 | 0.250 |
| $s^2 = 0.7$ | dCov | 0.952 | 0.912 | 0.640 |

**Example 3: SPD Manifold** $\mathcal{P}_d$. For the manifold of $d \times d$ SPD matrices $\mathcal{M} = \mathcal{P}_d$, we take $d = 2$ for computational convenience. The Fréchet mean is set as $\mu = I_2$. We define the fixed tangent vectors in $T_\mu \mathcal{P}_d$ as $v_1 = (1, 0.5, 1)^\top$ and $v_2 = (1, -0.5, 1)^\top$.

Results on $\mathcal{P}_d$ in Tables 5- 6 align with the patterns established on $\mathbb{S}^2$ and $\mathcal{W}_2(\mathbb{R})$. The MMDD-based test provides

*Table 4.* Rejection rates of MMDD-based test under DGP2, DGP3 ($s^2 = 0.5$), DGP4, DGP5 ($s^2 = 0.5$) on $\mathcal{W}_2(\mathbb{R})$.

| POWER | $\alpha = 0.1$ | | $\alpha = 0.05$ | | $\alpha = 0.01$ | |
|---|---|---|---|---|---|---|
| | $n = 100$ | $n = 200$ | $n = 100$ | $n = 200$ | $n = 100$ | $n = 200$ |
| DGP2 | | | | | | |
| $r^2 = 0.1$ | 0.110 | 0.176 | 0.044 | 0.079 | 0.005 | 0.017 |
| $r^2 = 0.2$ | 0.276 | 0.517 | 0.169 | 0.355 | 0.037 | 0.108 |
| $r^2 = 0.3$ | 0.565 | 0.912 | 0.397 | 0.834 | 0.108 | 0.333 |
| $r^2 = 0.4$ | 0.823 | 0.999 | 0.708 | 0.987 | 0.335 | 0.940 |
| $r^2 = 0.5$ | 0.947 | 1.000 | 0.887 | 1.000 | 0.595 | 0.998 |
| DGP3 | | | | | | |
| $r^2 = 0.1$ | 0.154 | 0.163 | 0.075 | 0.090 | 0.009 | 0.015 |
| $r^2 = 0.2$ | 0.250 | 0.411 | 0.147 | 0.290 | 0.021 | 0.079 |
| $r^2 = 0.3$ | 0.503 | 0.822 | 0.334 | 0.713 | 0.099 | 0.414 |
| $r^2 = 0.4$ | 0.740 | 0.986 | 0.598 | 0.963 | 0.279 | 0.829 |
| $r^2 = 0.5$ | 0.922 | 0.999 | 0.841 | 0.998 | 0.503 | 0.972 |
| DGP4 | | | | | | |
| $r^2 = 0.05$ | 0.145 | 0.187 | 0.074 | 0.110 | 0.005 | 0.015 |
| $r^2 = 0.10$ | 0.351 | 0.633 | 0.207 | 0.514 | 0.045 | 0.215 |
| $r^2 = 0.15$ | 0.645 | 0.954 | 0.495 | 0.921 | 0.187 | 0.751 |
| $r^2 = 0.20$ | 0.901 | 0.998 | 0.808 | 0.997 | 0.479 | 0.962 |
| DGP5 | | | | | | |
| $r^2 = 0.05$ | 0.149 | 0.177 | 0.068 | 0.095 | 0.005 | 0.018 |
| $r^2 = 0.10$ | 0.303 | 0.546 | 0.158 | 0.404 | 0.029 | 0.152 |
| $r^2 = 0.15$ | 0.587 | 0.912 | 0.442 | 0.835 | 0.145 | 0.594 |
| $r^2 = 0.20$ | 0.838 | 0.992 | 0.720 | 0.985 | 0.395 | 0.920 |

*Table 5.* Rejection rates of MMDD-based and dCov-based tests under DGP1 ($n = 100$) on $\mathcal{P}_d$.

| SIZE | | $\alpha = 0.1$ | $\alpha = 0.05$ | $\alpha = 0.01$ |
|---|---|---|---|---|
| $s^2 = 0$ | MMDD | 0.092 | 0.042 | 0.007 |
| $s^2 = 0.2$ | MMDD | 0.095 | 0.039 | 0.006 |
| $s^2 = 0.4$ | MMDD | 0.101 | 0.043 | 0.005 |
| $s^2 = 0.5$ | MMDD | 0.096 | 0.049 | 0.006 |
| $s^2 = 0.7$ | MMDD | 0.118 | 0.050 | 0.006 |
| $s^2 = 0$ | dCov | 0.097 | 0.049 | 0.008 |
| $s^2 = 0.2$ | dCov | 0.182 | 0.096 | 0.014 |
| $s^2 = 0.4$ | dCov | 0.572 | 0.394 | 0.152 |
| $s^2 = 0.5$ | dCov | 0.766 | 0.630 | 0.324 |
| $s^2 = 0.7$ | dCov | 0.975 | 0.932 | 0.724 |

*Table 6.* Rejection rates of MMDD-based test under DGP2, DGP3 ($s^2 = 0.5$), DGP4, DGP5 ($s^2 = 0.5$) on $\mathcal{P}_d$.

| POWER | $\alpha = 0.1$ | | $\alpha = 0.05$ | | $\alpha = 0.01$ | |
|---|---|---|---|---|---|---|
| | $n = 100$ | $n = 200$ | $n = 100$ | $n = 200$ | $n = 100$ | $n = 200$ |
| DGP2 | | | | | | |
| $r^2 = 0.1$ | 0.131 | 0.153 | 0.063 | 0.092 | 0.008 | 0.011 |
| $r^2 = 0.2$ | 0.273 | 0.445 | 0.165 | 0.316 | 0.032 | 0.115 |
| $r^2 = 0.3$ | 0.502 | 0.841 | 0.362 | 0.732 | 0.085 | 0.435 |
| $r^2 = 0.4$ | 0.745 | 0.987 | 0.607 | 0.972 | 0.245 | 0.858 |
| $r^2 = 0.5$ | 0.908 | 0.998 | 0.823 | 0.995 | 0.483 | 0.976 |
| DGP3 | | | | | | |
| $r^2 = 0.1$ | 0.143 | 0.160 | 0.071 | 0.078 | 0.010 | 0.014 |
| $r^2 = 0.2$ | 0.245 | 0.407 | 0.138 | 0.247 | 0.014 | 0.072 |
| $r^2 = 0.3$ | 0.447 | 0.744 | 0.312 | 0.622 | 0.083 | 0.319 |
| $r^2 = 0.4$ | 0.681 | 0.961 | 0.542 | 0.920 | 0.214 | 0.728 |
| $r^2 = 0.5$ | 0.857 | 0.996 | 0.747 | 0.986 | 0.401 | 0.934 |
| DGP4 | | | | | | |
| $r^2 = 0.05$ | 0.138 | 0.166 | 0.064 | 0.080 | 0.009 | 0.010 |
| $r^2 = 0.10$ | 0.279 | 0.587 | 0.169 | 0.444 | 0.028 | 0.185 |
| $r^2 = 0.15$ | 0.605 | 0.916 | 0.448 | 0.862 | 0.129 | 0.608 |
| $r^2 = 0.20$ | 0.843 | 0.994 | 0.728 | 0.984 | 0.370 | 0.911 |
| DGP5 | | | | | | |
| $r^2 = 0.05$ | 0.144 | 0.178 | 0.066 | 0.102 | 0.004 | 0.014 |
| $r^2 = 0.10$ | 0.285 | 0.463 | 0.160 | 0.309 | 0.020 | 0.102 |
| $r^2 = 0.15$ | 0.536 | 0.831 | 0.382 | 0.739 | 0.108 | 0.450 |
| $r^2 = 0.20$ | 0.756 | 0.974 | 0.623 | 0.948 | 0.301 | 0.813 |

valid size control and robust power under variance dependency, in contrast to the dCov-based test's size distortion.

In conclusion, the MMDD-based test demonstrates robust size control and consistent power performance across all three manifolds. In DGP1, it accurately distinguishes conditional mean independence from variance dependence, whereas the dCov-based test fails to separate these two forms of dependence. Computationally, the MMDD-based test (with Wild Bootstrap) requires only one precomputation of the inner product matrix, leading to a complexity of $O(n^2 + B \cdot n)$. In contrast, the dCov-based test (with permutation testing) must recompute the distance covariance statistic for each permutation, resulting in a higher complexity of $O(n^2 + n_{\text{perm}} \cdot n^2)$.

## 5. Application

We validate the proposed MMDD-based test using wind direction data. The dataset, curated from NOAA archives, comprises 118 U.S. international airports with complete records for the spherical response (wind direction) and all ten covariates. These covariates span geographic (AL-BERS_X, ALBERS_Y: airports' coordinates under Albers projection; ELEVATION: airports' elevation), meteorological (AWND: daily average wind speed; WSF2: maximum 2-minute wind speed; PRCP: daily precipitation; TMAX: daily maximum temperature; TMIN: daily minimum temperature), and temporal (PGTM: time of daily maximum atmospheric pressure, processed into periodic sine/cosine components to capture diurnal cyclicality) factors.

MMDD is designed to identify covariates that influence the conditional mean of a response—the primary target of regression analysis. Its practical advantage is illustrated by comparing it with distance covariance (dCov) on meteorological data (Figure 1). MMDD consistently ranks geographic factors (ALBERS_X, ALBERS_Y, ELEVATION) highest, whereas dCov elevates wind-speed indicators (WSF2, AWND). This discrepancy stems from their different targets: dCov captures global distributional dependence, so it detects the association between wind speed and the variability of wind direction. In contrast, MMDD specifically measures conditional mean dependence, thus

highlighting factors that govern the average wind direction—namely, geographic position and elevation, which physically determine regional wind regimes through large-scale circulation and topographic steering. Consequently, MMDD provides a more relevant screening tool for regression modeling by prioritizing covariates that directly affect the mean response.

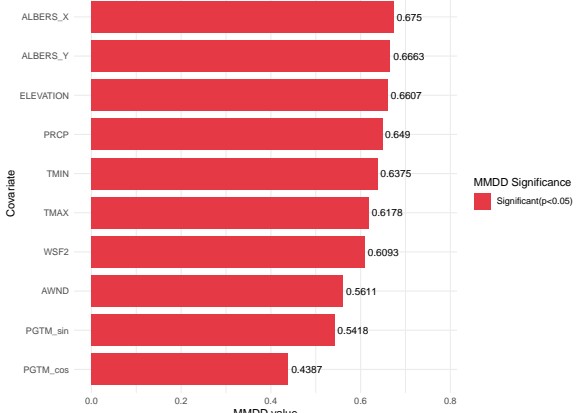

*(a)* MMDD-based covariate impact ranking on wind direction

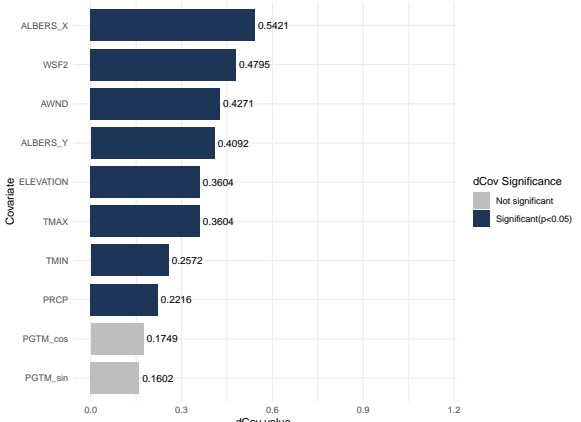

*(b)* dCov-based covariate impact ranking on wind direction

*Figure 1.* Covariate impact ranking comparisons between MMDD and dCov: (a) Displays the impact strength of covariates on wind direction (measured by MMDD, all significant at $p < 0.05$); (b) Presents the global dependence strength ranking (measured by dCov, with PGTM components non-significant).

A particularly illustrative contrast emerges for the diurnal covariate PGTM. MMDD detects a significant conditional mean dependence ($p < 0.05$), while dCov indicates no significant global dependence ($p > 0.05$). This finding is meteorologically coherent: PGTM modulates surface wind direction through diurnal pressure variations that drive local circulations (e.g., sea breezes). These processes systematically shift the average wind direction at specific times of day–a conditional mean effect that MMDD is designed to capture. The same signal, however, is diluted when data are aggregated across all times of day, as the systematic diurnal shifts become embedded within the broader, high-variability wind regime, leading dCov to overlook this physically meaningful, regression-relevant association.

Using the MMDD screening results, we fit a Fréchet regression model for wind-direction data ($Y \in \mathbb{S}^2$) following the Nadaraya-Watson (NW) estimator for random objects (Petersen & Müller, 2019). The model is specified as

$$Y_i = \exp_{m(X_i)}(\epsilon_i),$$

where $m(X)$ denotes the conditional Fréchet mean of $Y$ given $X$, $\exp_{m(X_i)}$ is the exponential map at $m(X_i)$, and $\epsilon_i \in T_{m(X_i)}\mathbb{S}^2$ is a tangent-space error term satisfying $\mathbb{E}[\epsilon_i|X_i] = 0$. The conditional Fréchet mean is estimated by minimizing the weighted squared geodesic distance

$$\hat{m}(x) = \arg\min_{\omega \in \mathbb{S}^2} \frac{1}{n} \sum_{i=1}^{n} K_h(X_i - x)\, d^2(Y_i, \omega),$$

with the Epanechnikov kernel $K_h$ and bandwidth $h = 0.2$. Tangent-space residuals are obtained via the logarithmic map

$$\hat{\epsilon}_i = \text{Log}_{\hat{m}(X_i)}(Y_i) \in T_{\hat{m}(X_i)}\mathbb{S}^2.$$

To assess model adequacy, we test the conditional mean independence of the error term. The hypotheses are

$$H_0 : \mathbb{E}(\epsilon|X) = 0 \text{ a.s.} \quad \text{vs} \quad H_1 : \Pr\{\mathbb{E}(\epsilon|X) \neq 0\} > 0.$$

We compute our test statistic as

$$T_n = n \cdot \frac{1}{n(n-1)} \sum_{i \neq j} \langle \hat{\epsilon}_i,\ \hat{\epsilon}_j \rangle_{T_{\hat{m}(X)}\mathbb{S}^2} k_{i,j}.$$

The p-value of the MMDD-based test is 0.956, leading to the acceptance of $H_0$. This directly validates the model's core assumption that the error term $\epsilon$ satisfies $\mathbb{E}(\epsilon|X) = 0$ a.s., confirming no systematic conditional mean dependence between the tangent-space residuals and covariates $X$. Thus, the NW-Fréchet regression captures the dominant conditional-mean dependence, while residual variation (likely due to unmeasured factors such as micro-topography) is stochastic and does not violate the conditional mean independence of the error term.

In summary, this application highlights two principal strengths of MMDD for analyzing manifold-valued meteorological data. First, in the important variable identification, it successfully prioritizes covariates with strong physical interpretations for the mean response, yielding a ranking consistent with atmospheric science. Second, it serves as a sensitive diagnostic tool for model adequacy by directly testing for residual conditional mean dependence. Compared to omnibus tests like dCov, MMDD provides a more interpretable framework for supporting regression analysis when modeling the average outcome is the primary goal.

# 6. Conclusion and Discussion

This paper proposes the Manifold Martingale Difference Divergence (MMDD), a nonparametric test for conditional mean independence between Euclidean predictors and manifold-valued responses. By projecting data onto the tangent space at the Fréchet mean via the logarithmic map, MMDD isolates regression-relevant conditional mean dependence, unlike distance covariance, which targets global distributional coupling. Simulations on three representative manifolds confirm that MMDD maintains the correct size under the null, even in the presence of marginal distribution dependence, while achieving competitive power across linear and nonlinear alternatives. In practice, MMDD provides two immediate utilities for machine learning workflows. First, it enables covariate screening prior to training geometric neural networks (e.g., spherical or SPD networks) by identifying which variables drive the conditional mean of the manifold response—an essential step for building interpretable, parsimonious models. Second, it serves as a post-hoc diagnostic for any regression or generative model on manifolds: after fitting, testing whether residuals still exhibit conditional mean dependence on the covariates reveals potential model misspecification. These capabilities make MMDD a practical tool for robust, data-driven analysis of non-Euclidean data.

One promising direction for future research is to extend the MMDD framework to test conditional quantile independence for manifold-valued responses. Unlike mean independence, conditional quantile independence captures more nuanced and localized dependence structures, offering a richer characterization of the relationship between covariates and non-Euclidean outcomes. Existing theoretical links between mean and quantile independence in Euclidean settings, such as the martingale-based approach of Shao & Zhang (2014), could inform the adaptation of MMDD to manifold-valued quantiles. For instance, manifold quantiles may be defined via spatial depth or other depth-based notions tailored to non-Euclidean data, leading to a rigorous and interpretable test of conditional quantile independence on manifolds.

## Acknowledgements

Dr. You's research was supported by the 111-Center Project of China (No. B25066) and the Innovative Research Team of Shanghai University of Finance and Economics. Dr. Wang's research was supported by the National Natural Science Foundation of China (No.72501165).

## Impact Statement

This paper presents work whose goal is to advance the field of Machine Learning. There are many potential societal consequences of our work, none which we feel must be specifically highlighted here.

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

# A. Appendix

## A.1. Equivalence between Conditional Fréchet Mean and Tangent Space Conditional Expectation

We provide an intuitive explanation for the equivalence

$$\mathbb{E}(Y|X) = \mathbb{E}(Y) = \mu \quad \Longleftrightarrow \quad \mathbb{E}\big(\text{Log}_\mu(Y)|X\big) = 0,$$

which was used in the main text without formal presentation as a proposition.

Let $\phi_x(\omega) = \mathbb{E}[d(Y,\omega)^2|X=x]$ be the conditional Fréchet functional, with minimizer $m_x$ (the conditional Fréchet mean given $X=x$). If $m_x = \mu$ for almost all $x$, then $\mu$ minimizes $\phi_x$ for each $x$. Under appropriate regularity conditions (differentiability and exchangeability), the first-order optimality condition at $\mu$ yields

$$\nabla\phi_x(\mu) = -2\,\mathbb{E}\big[\text{Log}_\mu(Y)\,|\,X=x\big] = 0,$$

implying $\mathbb{E}[\text{Log}_\mu(Y)|X=x] = 0$.

Conversely, if $\mathbb{E}[\text{Log}_\mu(Y)|X=x] = 0$, using the local quadratic approximation of the squared geodesic distance near $\mu$,

$$d^2(Y, \exp_\mu(v)) \approx \|\text{Log}_\mu(Y) - v\|^2 \quad \text{for small } \|v\|,$$

we have

$$\phi_x(\exp_\mu(v)) \approx \mathbb{E}\big[\|\text{Log}_\mu(Y) - v\|^2\,|\,X=x\big].$$

The right-hand side is minimized at $v = 0$, suggesting that $\mu$ is a local minimizer of $\phi_x$. Under uniqueness of the conditional Fréchet mean, we obtain $m_x = \mu$.

## A.2. Proof of Proposition 2.3

*Proof.* Recall the definition of the MMDD statistic

$$\text{MMDD}(Y|X) = \mathbb{E}\{\langle \text{Log}_\mu(Y), \text{Log}_\mu(Y')\rangle_{T_\mu\mathcal{M}} \cdot K(X - X')\}.$$

We next prove the two stated properties.

(1) The inner product on the tangent space $T_\mu\mathcal{M}$ is positive semi-definite, yielding $\langle \text{Log}_\mu(Y), \text{Log}_\mu(Y')\rangle_{T_\mu\mathcal{M}} \geq 0$. Since $K$ is a positive semi-definite kernel, we have that $K(X - X') \geq 0$ for all $X, X' \in \mathbb{R}^p$. Consequently, their product is non-negative, and taking the expectation preserves this inequality $\text{MMDD}(Y|X) \geq 0$.

(2) We demonstrate that $\text{MMDD}(Y|X) = 0$ if and only if $H_0 : \mathbb{E}(\text{Log}_\mu(Y)|X) = 0$ holds. The justification follows from properties of conditional expectation and Fourier analysis:

$$\begin{aligned}
\mathbb{E}(\text{Log}_\mu(Y)|X) = 0 &\Longleftrightarrow \mathbb{E}\left\{\text{Log}_\mu(Y)\,e^{i\langle t,X\rangle}\right\} = 0, \quad \forall t \in \mathbb{R}^p \\
&\Longleftrightarrow \left\langle \mathbb{E}\left\{\text{Log}_\mu(Y)e^{i\langle t,X\rangle}\right\}, \mathbb{E}\left\{\text{Log}_\mu(Y)e^{i\langle t,X\rangle}\right\}\right\rangle_{T_\mu\mathcal{M}} = 0, \quad \forall t \in \mathbb{R}^p \\
&\Longleftrightarrow \mathbb{E}\left\{\langle \text{Log}_\mu(Y), \text{Log}_\mu(Y')\rangle_{T_\mu\mathcal{M}}\, e^{i\langle t,X\rangle}e^{-i\langle t,X'\rangle}\right\} = 0, \quad \forall t \in \mathbb{R}^p \\
&\Longleftrightarrow \int_{\mathbb{R}^p} \mathbb{E}\left\{\langle \text{Log}_\mu(Y), \text{Log}_\mu(Y')\rangle_{T_\mu\mathcal{M}}\, e^{i\langle t,X-X'\rangle}\right\}\,dw(t) = 0 \\
&\Longleftrightarrow \mathbb{E}\left\{\langle \text{Log}_\mu(Y), \text{Log}_\mu(Y')\rangle_{T_\mu\mathcal{M}} \int_{\mathbb{R}^p} e^{i\langle t,X-X'\rangle}dw(t)\right\} = 0 \\
&\Longleftrightarrow \mathbb{E}\left\{\langle \text{Log}_\mu(Y), \text{Log}_\mu(Y')\rangle_{T_\mu\mathcal{M}} \cdot K(X - X')\right\} = 0 \\
&\Longleftrightarrow \text{MMDD}(Y|X) = 0.
\end{aligned}$$

The final equivalence employs Bochner's theorem of Bochner (1959), which guarantees that $K(z) = \int_{\mathbb{R}^p} e^{-iz^\top t}dw(t)$ is a positive semi-definite function corresponding to a finite nonnegative Borel measure $w$. This completes the proof. $\square$

### A.3. Proof of Theorem 2.6

*Proof.* The empirical statistic $\mathrm{MMDD}_n(Y|X)$ can be expressed as

$$\mathrm{MMDD}_n(Y|X) = \frac{1}{n(n-1)} \sum_{i \neq j} H(\hat{\mu}; W_i, W_j),$$

where $W_i = (X_i, Y_i)$ and

$$H(\hat{\mu}; W_i, W_j) := \langle \mathrm{Log}_{\hat{\mu}}(Y_i), \mathrm{Log}_{\hat{\mu}}(Y_j) \rangle_{T_{\hat{\mu}}\mathcal{M}} K(X_i - X_j).$$

Define the tangent space projections $\xi_i := \mathrm{Log}_\mu(Y_i) \in T_\mu\mathcal{M}$ and $\xi_{\hat{\mu}} := \mathrm{Log}_\mu(\hat{\mu})$, where $\xi_{\hat{\mu}}$ represents the empirical estimation error of the Fréchet mean. Under regularity conditions, we have $\xi_{\hat{\mu}} = O_p(n^{-1/2})$.

To analyze the asymptotic behavior, we perform a second-order Taylor expansion of $H(\hat{\mu}; W_i, W_j)$ around the true Fréchet mean $\mu$:

$$H(\hat{\mu}; W_i, W_j) = H(\mu; W_i, W_j) + \langle g_{i,j}, \xi_{\hat{\mu}} \rangle_{T_\mu\mathcal{M}} + \frac{1}{2}\langle \mathcal{H}_{i,j}\xi_{\hat{\mu}}, \xi_{\hat{\mu}} \rangle_{T_\mu\mathcal{M}} + o(\|\xi_{\hat{\mu}}\|^2_{T_\mu\mathcal{M}}),$$

where

$$g_{i,j} := \nabla_\mu H(\mu; W_i, W_j) \quad \text{and} \quad \mathcal{H}_{i,j} := \nabla^2_\mu H(\mu; W_i, W_j)$$

denote the Riemannian gradient and Riemannian Hessian operator, respectively, both evaluated at $\mu$.

Recalling that $T_n = n\mathrm{MMDD}_n(Y|X)$, we decompose $T_n$ as follows

$$T_n = \frac{1}{n-1}\sum_{i \neq j} H(\mu; W_i, W_j) + \frac{1}{n-1}\sum_{i \neq j}\langle g_{i,j}, \xi_{\hat{\mu}} \rangle + \frac{1}{2(n-1)}\sum_{i \neq j}\langle \mathcal{H}_{i,j}\xi_{\hat{\mu}}, \xi_{\hat{\mu}} \rangle + o_p(1)$$

$$= T_{n1} + T_{n2} + T_{n3} + o_p(1). \tag{10}$$

Observe that after the Taylor expansion at $\mu$, the original manifold-valued responses $Y_i$ appear only through their tangent space projections $\xi_i = \mathrm{Log}_\mu(Y_i)$. In particular,

$$H(\mu; W_i, W_j) = \langle \xi_i, \xi_j \rangle_{T_\mu\mathcal{M}} K(X_i - X_j),$$

and the gradient $g_{i,j}$ and Hessian operator $\mathcal{H}_{i,j}$, being derivatives of $H$ at $\mu$, also depend on $W_i, W_j$ only via $\xi_i, \xi_j$ and $X_i, X_j$. Hence, each term $T_{nm}$ ($m = 1, 2, 3$) in the decomposition is a function of the pairs $(X_i, \xi_i)$.

For notational convenience, we therefore define the combined random vectors

$$Z_i := (X_i, \xi_i) \in \mathbb{R}^p \times T_\mu\mathcal{M}, \qquad i = 1, \ldots, n.$$

Lemmas A.1-A.3 below establish the asymptotic properties of $T_{nm}, m = 1, 2, 3$. Combining the results in Lemmas A.1-A.3, we have

$$T_n - B_1 = n\binom{n}{2}^{-1}\sum_{i<j}\sum_{m=1}^3 h_m(Z_i, Z_j) + o_p(1),$$

where each $h_m$ is the kernel function of a second-order degenerate U-statistic. The term $n\binom{n}{2}^{-1}\sum_{i<j}\sum_{m=1}^3 h_m(Z_i, Z_j)$ is a second-order degenerate U-statistic with kernel

$$h(Z_i, Z_j) := \sum_{m=1}^3 h_m(Z_i, Z_j)$$

$$= \langle \xi_i, \xi_j \rangle_{T_\mu\mathcal{M}} K(X_i - X_j) + \langle \mathbb{E}_1(g_{i,1}), A^{-1}\xi_j \rangle + \langle \mathbb{E}_1(g_{j,1}), A^{-1}\xi_i \rangle + \frac{1}{2}\langle \mathbb{E}(\mathcal{H}_{1,2}) A^{-1}\xi_i, A^{-1}\xi_j \rangle. \tag{11}$$

By Dunford et al. (1963, p.1087), and Theorems 2.1 and 2.3 in Gregory (1977),

$$T_n - B_1 \xrightarrow{d} \sum_{\nu=1}^\infty \lambda_\nu(G_\nu^2 - 1) \quad \text{and} \quad T_n \xrightarrow{d} \sum_{\nu=1}^\infty \lambda_\nu G_\nu^2,$$

where $B_1 = \sum_{m=1}^3 \mathbb{E}[h_m(Z_i, Z_i)]$, $G_\nu \overset{i.i.d.}{\sim} \mathcal{N}(0, 1)$, and $\lambda_\nu$ are eigenvalues of the integral equation:

$$\int_{-\infty}^\infty \sum_{m=1}^3 h_m(Z_1, Z_2) f_\nu(Z_2) dF(Z_2) = \lambda_\nu f_\nu(Z_1).$$

$\square$

**Lemma A.1.** $T_{n1} = n\binom{n}{2}^{-1} \sum_{i<j} h_1(Z_i, Z_j)$, where

$$h_1(Z_i, Z_j) = \langle \xi_i, \xi_j \rangle_{T_\mu \mathcal{M}} K(X_i - X_j)$$

*Proof.* $T_{n1} = nU_{n1}$, where $U_{n1} = \binom{n}{2}^{-1} \sum_{i<j} \psi^{(1)}(Z_i, Z_j)$ and

$$\psi^{(1)}(Z_i, Z_j) = \langle \xi_i, \xi_j \rangle_{T_\mu \mathcal{M}} K(X_i - X_j).$$

By the Cauchy-Schwarz inequality, we have that $K(x_1 - x_2)$ is uniformly bounded. This, together with the fact that $\psi^{(1)}$ is symmetric in its two arguments, yields that $\mathbb{E}_1\big[\psi^{(1)}(Z_1, Z_2)\big|Z_2\big] = 0$ and $\mathbb{E}\big[\psi^{(1)}(Z_1, Z_2)^2\big] < \infty$. Thus, $U_{n1}$ is a second order degenerate U-statistic satisfying the conditions of Theorem 1 in Section 3.2.2 of Lee (2019, pp. 79–80). It follows that

$$nU_{n1} \overset{d}{\to} \sum_{\nu=1}^\infty \lambda_\nu^{(1)}(G_\nu^2 - 1),$$

where $G_\nu$'s are i.i.d. $\mathcal{N}(0, 1)$ and $\lambda_\nu^{(1)}$'s are the eigenvalues of the integral equation

$$\int \psi^{(1)}(Z_1, Z_2) f(Z_2) \, dF(Z_2) = \lambda f(Z_1)$$

with $f$ and $F$ being the probability density function and limiting cumulative distribution function of $Z_i$, respectively. $\square$

**Lemma A.2.** $T_{n2} - 2\mathbb{E}[\langle g_{1,2}, A^{-1}\xi_1 \rangle] = n\binom{n}{2}^{-1} \sum_{i<j} h_2(Z_i, Z_j) + o_p(1)$, where $h_2(Z_i, Z_j) = \langle \mathbb{E}_1(g_{i,1}), A^{-1}\xi_j \rangle + \langle \mathbb{E}_1(g_{j,1}), A^{-1}\xi_i \rangle$.

*Proof.* By Assumption 2.5, it follows that $\xi_{\hat\mu} = A^{-1}\frac{1}{n}\sum_{k=1}^n \xi_k + o_p(n^{-1/2})$. Then we have

$$T_{n2} = \frac{1}{n(n-1)} \sum_{i \neq j} \sum_{k=1}^n \langle g_{i,j}, A^{-1}\xi_k \rangle + o_p(1) = \bar{T}_{n2} + o_p(1).$$

Next, we make the following decomposition:

$$\bar{T}_{n2} = \frac{1}{n(n-1)} \sum_{i \neq j} \langle g_{i,j}, A^{-1}\xi_i \rangle + \frac{1}{n(n-1)} \sum_{i \neq j} \langle g_{i,j}, A^{-1}\xi_j \rangle + \frac{1}{n(n-1)} \sum_{i \neq j \neq k} \langle g_{i,j}, A^{-1}\xi_k \rangle$$

$$= T_{n2,1} + T_{n2,2} + T_{n2,3}.$$

By the law of large numbers (WLLN) for the second order U-statistic, we can readily show that $T_{n2,1} = \mathbb{E}\big[\langle g_{1,2}, A^{-1}\xi_1 \rangle\big] + o_p(1)$ and $T_{n2,2} = \mathbb{E}\big[\langle g_{1,2}, A^{-1}\xi_2 \rangle\big] + o_p(1)$. Next, notice that $T_{n2,3} = \frac{n-2}{n}nU_{n2}$, where $U_{n2} = \binom{n}{3}^{-1} \sum_{i<j<k} \psi^{(2)}(Z_i, Z_j, Z_k)$ and $\psi^{(2)}(Z_i, Z_j, Z_k) = \frac{1}{3}\big(\langle g_{i,j}, A^{-1}\xi_k \rangle + \langle g_{i,k}, A^{-1}\xi_j \rangle + \langle g_{j,k}, A^{-1}\xi_i \rangle\big)$. Noting that $\psi^{(2)}$ is symmetric in its three arguments, $\mathbb{E}[\psi^{(2)}(Z_1, Z_2, Z_3)] = 0$, $\mathbb{E}[\psi^{(2)}(Z_1, Z_2, Z_3)|Z_1] = 0$, and $\mathbb{E}[\psi^{(2)}(Z_1, Z_2, Z_3)|Z_1, Z_2] = \frac{1}{3}\big(\langle \mathbb{E}_3(g_{1,3}), A^{-1}\xi_2 \rangle + \langle \mathbb{E}_3(g_{2,3}), A^{-1}\xi_1 \rangle\big) := h_2^{(2)}(Z_1, Z_2)$. Let $h_3^{(2)}(Z_1, Z_2, Z_3) = \psi^{(2)}(Z_1, Z_2, Z_3) - \big(h_2^{(2)}(Z_1, Z_2) + h_2^{(2)}(Z_1, Z_3) + h_2^{(2)}(Z_2, Z_3)\big)$. By Hoeffding decomposition (Lee, 2019, p.26), we have

$$U_{n2} = 3H_{2n}^{(2)} + H_{3n}^{(2)},$$

where $H_{2n}^{(2)} = \binom{n}{2}^{-1} \sum_{i<j} h_2^{(2)}(Z_i, Z_j)$ and $H_{3n}^{(2)} = \binom{n}{3}^{-1} \sum_{i<j<k} h_3^{(2)}(Z_i, Z_j, Z_k)$. By moment calculations, $\mathbb{E}[H_{3n}^{(2)}] = 0$ and $\mathrm{Var}[H_{3n}^{(2)}] = O(n^{-3})$, implying that $H_{3n}^{(2)} = O_p(n^{-3/2})$. In addition, $H_{2n}^{(2)}$ is a standard second order degenerate U-statistic such that $nH_{2n}^{(2)} = O_p(1)$. It follows that

$$T_{n2,3} = n\binom{n}{2}^{-1} \sum_{i<j} h_2(Z_i, Z_j) + O_p(n^{-1/2}),$$

where $h_2(Z_i, Z_j) = \langle \mathbb{E}_1(g_{i,1}), A^{-1}\xi_j \rangle + \langle \mathbb{E}_1(g_{j,1}), A^{-1}\xi_i \rangle$. Combining these results, we have $T_{n2} - 2\mathbb{E}[\langle g_{1,2}, A^{-1}\xi_1 \rangle] = n\binom{n}{2}^{-1} \sum_{i<j} h_2(Z_i, Z_j) + o_p(1)$. $\square$

**Lemma A.3.** $T_{n3} - \frac{1}{2}\mathbb{E}[\langle \mathcal{H}_{1,2}A^{-1}\xi_3, A^{-1}\xi_3 \rangle] = n\binom{n}{2}^{-1} \sum_{i<j} h_3(Z_i, Z_j) + o_p(1)$, where $h_3(Z_i, Z_j) = \frac{1}{2}\langle \mathbb{E}(\mathcal{H}_{1,2}) A^{-1}\xi_i, A^{-1}\xi_j \rangle$.

*Proof.* By Assumption 2.5, we can readily show that $T_{n3} = \bar{T}_{n3} + o_p(1)$, where

$$\bar{T}_{n3} = \frac{1}{2n^2(n-1)} \sum_{i \neq j} \sum_{k=1}^{n} \sum_{l=1}^{n} \langle \mathcal{H}_{i,j} A^{-1}\xi_k, A^{-1}\xi_l \rangle$$

$$= \frac{1}{2n^2(n-1)} \sum_{i \neq j \neq k \neq l} \langle \mathcal{H}_{i,j} A^{-1}\xi_k, A^{-1}\xi_l \rangle + \frac{1}{2n^2(n-1)} \sum_{i \neq j \neq k} \langle \mathcal{H}_{i,j} A^{-1}\xi_k, A^{-1}\xi_k \rangle + o_p(1)$$

$$= T_{n3,1} + T_{n3,2} + o_p(1).$$

Then we write $T_{n3,1} = \frac{(n-2)(n-3)}{n^2} nU_{n3}$, where $U_{n3} := \binom{n}{4}^{-1} \sum_{i<j<k<\ell} \psi^{(3)}(Z_i, Z_j, Z_k, Z_\ell)$, $\psi^{(3)}(Z_i, Z_j, Z_k, Z_\ell) = \frac{1}{48} \sum_{4!} \langle \mathcal{H}_{i,j} A^{-1}\xi_k, A^{-1}\xi_l \rangle$, and $\sum_{4!}$ denotes the summation over all the 4! kinds of permutation of $\{i, j, k, l\}$. By Hoeffding decomposition, we can readily show that $U_{n3} = \binom{n}{2}^{-1} \sum_{i<j} h_3(Z_i, Z_j) + O_p(n^{-3/2})$, where $h_3(Z_i, Z_j) = \frac{1}{2}\langle \mathbb{E}(\mathcal{H}_{1,2}) A^{-1}\xi_i, A^{-1}\xi_j \rangle$. In addition, by WLLN for U-statistics, we have $T_{n3,2} = \frac{1}{2}\mathbb{E}[\langle \mathcal{H}_{12}A^{-1}\xi_3, A^{-1}\xi_3 \rangle] + o_p(1)$. It follows that

$$T_{n3} - \frac{1}{2}\mathbb{E}[\langle \mathcal{H}_{1,2}A^{-1}\xi_3, A^{-1}\xi_3 \rangle] = n\binom{n}{2}^{-1} \sum_{i<j} h_3(Z_i, Z_j) + o_p(1).$$

$\square$

## A.4. Proof of Theorem 2.7

*Proof.* Under the local alternative $H_{1,n} : \mathrm{Log}_\mu(Y) = n^{-a}g(X) + \varepsilon$, we have $\langle \mathrm{Log}_\mu(Y_i), \mathrm{Log}_\mu(Y_j) \rangle = n^{-2a}\langle g(X_i), g(X_j) \rangle + n^{-a} (\langle g(X_i), \varepsilon_j \rangle + \langle \varepsilon_i, g(X_j) \rangle) + \langle \varepsilon_i, \varepsilon_j \rangle$. Let $k_{i,j} := K(X_i - X_j)$ and $Z_i = (X_i, \varepsilon_i)$. From (10), it follows that

$$T_{n1} = \frac{1}{n-1} \sum_{i \neq j} \langle \mathrm{Log}_\mu(Y_i), \mathrm{Log}_\mu(Y_j) \rangle k_{i,j}$$

$$= \frac{1}{n-1} \sum_{i \neq j} \left\{ n^{-2a}\langle g(X_i), g(X_j) \rangle k_{i,j} + n^{-a} (\langle g(X_i), \varepsilon_j \rangle + \langle g(X_j), \varepsilon_i \rangle) k_{i,j} + \langle \varepsilon_i, \varepsilon_j \rangle k_{i,j} \right\}$$

$$= T_{n1,1} + T_{n1,2} + T_{n1,3}.$$

**Case 1:** $0 < a < 1/2$. We write $T_{n1,1} = n^{-2a} \times nU_{n1}$, where $U_{n1} = \binom{n}{2}^{-1} \sum_{i<j} \phi^{(1)}(X_i, X_j)$ and $\phi^{(1)}(X_i, X_j) = \langle g(X_i), g(X_j) \rangle k_{i,j}$. By WLLN for U-statistics, we can readily show that $U_{n1} = \mathbb{E}(\phi^{(1)}(X_1, X_2)) + o_p(1) = O_p(1)$, since $\mathbb{E}(\phi^{(1)}(X_1, X_2)) > 0$. If $0 < a < 1/2$, then $T_{n1,1} = n^{1-2a}O_p(1) \xrightarrow{p} \infty$ and thus $T_n \xrightarrow{p} \infty$.

**Case 2:** $a = 1/2$. We write $T_{n1,2} = n^{-1/2} \times nU_{n2}$, where $U_{n2} = \binom{n}{2}^{-1} \sum_{i<j} \phi^{(2)}(Z_i, Z_j)$ and $\phi^{(2)}(Z_i, Z_j) = \left( \langle g(X_i), \varepsilon_j \rangle + \langle g(X_j), \varepsilon_i \rangle \right) k_{i,j}$. Noting that $\phi^{(2)}$ is symmetric in its two arguments, and $\mathbb{E}_1\left[ \phi^{(2)}(Z_1, Z_2) | Z_2 \right] = 0$, $\mathbb{E}\left[ \phi^{(2)}(Z_1, Z_2)^2 \right] < \infty$. So $U_{n2}$ is a second order degenerate U-statistic such that $nU_{n2} = O_p(1)$. Then we have $T_{n1,2} = O_p(n^{-1/2})$. Also, we write $T_{n1,3} = nU_{n3}$, where $U_{n3} = \binom{n}{2}^{-1} \sum_{i<j} \phi^{(3)}(Z_i, Z_j)$ and $\phi^{(3)}(Z_i, Z_j) = \langle \varepsilon_i, \varepsilon_j \rangle k_{i,j}$. In addition, $U_{n3}$ is a standard second order degenerate U-statistic. Then we have

$$T_{n1} - \mathbb{E}[\langle g(X_1), g(X_2) \rangle k_{1,2}] = n \binom{n}{2}^{-1} \sum_{i<j} h_1(Z_i, Z_j) + o_p(1) \tag{12}$$

where $h_1(Z_i, Z_j) = \langle \varepsilon_i, \varepsilon_j \rangle k_{i,j}$. Note that the empirical Fréchet estimator $\hat{\mu}$ is also $\sqrt{n}$-consistent under $H_{1,n}$, we can readily follow the proof of Lemma A.2-A.3 and show that

$$T_{n2} - 2\mathbb{E}[\langle g_{1,2}, A^{-1}\varepsilon_1 \rangle] = n \binom{n}{2}^{-1} \sum_{i<j} h_2(Z_i, Z_j) + o_p(1), \tag{13}$$

$$T_{n3} - \frac{1}{2}\mathbb{E}\left[ \langle \mathcal{H}_{1,2} A^{-1}\varepsilon_3, \ A^{-1}\varepsilon_3 \rangle \right] = n \binom{n}{2}^{-1} \sum_{i<j} h_3(Z_i, Z_j) + o_p(1), \tag{14}$$

where $h_2(Z_i, Z_j) = \langle \mathbb{E}_1(g_{i,1}), A^{-1}\varepsilon_i \rangle + \langle \mathbb{E}_1(g_{j,1}), A^{-1}\varepsilon_j \rangle$ and $h_3(Z_i, Z_j) = \frac{1}{2} \langle \mathbb{E}(\mathcal{H}_{1,2}) A^{-1}\varepsilon_i, \ A^{-1}\varepsilon_j \rangle$. Combining the results, we have

$$T_n - B_2 = n \binom{n}{2}^{-1} \sum_{i<j} \sum_{m=1}^{3} h_m(Z_i, Z_j) + o_p(1),$$

where $B_2 = B_1 + \mathbb{E}[\langle g(X_1), g(X_2) \rangle k_{1,2}]$ and $B_1 = \sum_{m=1}^{3} \mathbb{E}[h_m(Z_i, Z_i)]$. The conclusion then follows from Theorem 2.3 in Gregory (1977).

**Case 3:** $a > 1/2$. $T_{n1} = n\binom{n}{2}^{-1} \sum_{i<j} h_1(Z_i, Z_j) + o_p(1)$, $T_{n,2}$ and $T_{n,3}$ are the same as (13) and (14). Then we have

$$T_n - B_1 = n \binom{n}{2}^{-1} \sum_{i<j} \sum_{m=1}^{3} h_m(Z_i, Z_j) + o_p(1).$$

As a direct application of Theorem 2.3 from Gregory (1977), the conclusion follows.

$\square$

### A.5. Proof of Theorem 3.3

*Proof.* Recall the wild bootstrap statistic for the MMDD test

$$T_n^* = \frac{1}{n-1} \sum_{i \neq j} \eta_i \eta_j \left\langle \mathrm{Log}_{\hat{\mu}}(Y_i), \mathrm{Log}_{\hat{\mu}}(Y_j) \right\rangle_{T_{\hat{\mu}}\mathcal{M}} k_{i,j}.$$

Follow the proof of Lemmas A.1-A.3, we can readily show that

$$T_n^* - B_1 = n \binom{n}{2}^{-1} \sum_{i<j} \sum_{m=1}^{3} h_m(Z_i, Z_j)\eta_i\eta_j + o_p(1) = nU_n^* + o_p(1),$$

where the kernels $h_m$ and $B_1$ are the same as in Section A.3. We first verify the conditions of Theorem 4 in Lee et al. (2020) for our framework:

1. Symmetric kernel: $h(Z_i, Z_j) = \sum_{m=1}^{3} h_m(Z_i, Z_j)$ is symmetric.

2. Finite fourth moment: $\mathbb{E}\left[ h(Z, Z')^4 \right] < \infty$ (follows from our Assumptions of Theorem 3.3).

3. Weight moment conditions: i.i.d. weights $\{\eta_i\}$ satisfy $\mathbb{E}(\eta_i) = 0$, $\text{Var}(\eta_i) = 1$, $\mathbb{E}(\eta_i^4) < \infty$.

By Theorem 4 of Lee et al. (2020), the bootstrap statistic $nU_n^* = \frac{1}{n-1}\sum_{i \neq j} h(Z_i, Z_j)\eta_i\eta_j$ has the asymptotic distribution

$$nU_n^* \overset{d^*}{\to} \sum_{\nu=1}^{\infty} \lambda_\nu(G_\nu^2 - 1) \quad \text{a.s.},$$

where $\{\lambda_\nu\}$ are eigenvalues of the integral operator induced by kernel $h$, and $\{G_\nu\}$ are i.i.d. standard normals.

Recall $T_n^* - B_1 = nU_n^* + o_p(1)$, then we have

$$T_n^* - B_1 \overset{d^*}{\to} \sum_{\nu=1}^{\infty} \lambda_\nu(G_\nu^2 - 1) \quad \text{a.s.,} \quad \text{and} \quad T_n^* \overset{d^*}{\to} \sum_{\nu=1}^{\infty} \lambda_\nu G_\nu^2 \quad \text{a.s.}$$

with $B_1 = \sum_{m=1}^{3} \mathbb{E}[h_m(Z_i, Z_i)]$. $\qquad\qquad\square$

### A.6. Proof of Theorem 3.4

*Proof.* We establish the asymptotic power of the MMDD test under the local alternative

$$H_{1,n} : \text{Log}_\mu(Y) = n^{-a}g(X) + \epsilon, \quad a > 0,$$

where $g : \mathbb{R}^p \to T_\mu\mathcal{M}$ satisfies $\mathbb{E}[g(X)] = 0$ and $\epsilon$ is independent of $X$ with $\mathbb{E}[\epsilon|X] = 0$.

**Step 1. Asymptotic behavior of $T_n$ under $H_{1,n}$.** From Theorem 2, under the local alternative $H_{1,n}$,

- **Case 1:** $0 < a < 1/2$. Then, we obtain $T_n \overset{p}{\to} \infty$.

- **Case 2:** $a = 1/2$. Then, we obtain

$$T_n \overset{d}{\to} c + \sum_{\nu=1}^{\infty} \lambda_\nu G_\nu^2, \ c > 0.$$

- **Case 3:** $a > 1/2$. Then, we obtain

$$T_n \overset{d}{\to} \sum_{\nu=1}^{\infty} \lambda_\nu G_\nu^2.$$

**Step 2. Asymptotic distribution of $T_n^*$ under $H_{1,n}$.** The bootstrap statistic is defined as

$$T_n^* = \frac{1}{n-1} \sum_{i \neq j} \eta_i\eta_j \ \langle \text{Log}_{\hat{\mu}}Y_i, \text{Log}_{\hat{\mu}}Y_j \rangle_{T_{\hat{\mu}}\mathcal{M}} \ k_{i,j}.$$

Following the decomposition in the proof of Section A.4, the bootstrap statistic $T_n^*$ is obtained by multiplying each term in the U-statistic by the external weights $\eta_i\eta_j$. In each case we can write

$$T_n^* - n^{1-2a}\mathbb{E}[\langle g(X_1), g(X_2)\rangle k_{1,2}\eta_1\eta_2] - B_1 = n\binom{n}{2}^{-1} \sum_{i<j} \sum_{m=1}^{3} h_m(Z_i, Z_j)\eta_i\eta_j + o_p^*(1),$$

where the kernels $h_m$ and $B_1$ are the same as in Section A.4. The expectation term $\mathbb{E}[\langle g(X_1), g(X_2)\rangle k_{1,2}\eta_1\eta_2] = 0$ since $\mathbb{E}(\eta_i) = 0$ and $\eta_i$ is independent of $X_i$. Thus, we have

$$T_n^* - B_1 = n\binom{n}{2}^{-1} \sum_{i<j} \sum_{m=1}^{3} h_m(Z_i, Z_j)\eta_i\eta_j + o_p^*(1),$$

Under the assumptions of Theorem 3.4, the conditions of Theorem 4 in Lee et al. (2020) are satisfied. Consequently, in all three cases, we obtain

$$T_n^* \overset{d^*}{\longrightarrow} \sum_{\nu=1}^{\infty} \lambda_\nu G_\nu^2 \quad \text{a.s.}$$

This is exactly the same limiting distribution as under $H_0$ (see Theorem 3.3). Consequently, the bootstrap critical value $Q_{(1-\alpha),n}^*$ converges in probability to the $(1-\alpha)$-quantile of this distribution, denoted by $Q_{(1-\alpha),\mathcal{G}_0}$.

**Step 3. Limiting power under** $H_{1,n}$**.** Now, combining the asymptotic behavior of $T_n$ (Step 1) and that of $Q^*_{(1-\alpha),n}$ (Step 2), we obtain:

- **Case 1:** $0 < a < 1/2$. Since $T_n \xrightarrow{p} \infty$ and $Q^*_{(1-\alpha),n}$ converges to a finite constant, we have $\Pr\{T_n \geq Q^*_{(1-\alpha),n}\} \to 1$.

- **Case 2:** $a = 1/2$. Here $T_n$ converges in distribution to a non-central weighted chi-squared variable: $T_n \xrightarrow{d} \sum_\nu \lambda_\nu (G_\nu + b_\nu)^2$. Meanwhile, $Q^*_{(1-\alpha),n} \xrightarrow{p} Q_{(1-\alpha),\mathcal{G}_0}$. Therefore,

$$\Pr\{T_n \geq Q^*_{(1-\alpha),n}\} \to \Pr\left\{\sum_\nu \lambda_\nu G_\nu^2 \geq Q_{(1-\alpha),\mathcal{G}_0} - c\right\}.$$

- **Case 3:** $a > 1/2$. In this case, $T_n$ has the same limit as under $H_0$, i.e., $T_n \xrightarrow{d} \sum_\nu \lambda_\nu G_\nu^2$. Consequently, $\Pr\{T_n \geq Q^*_{(1-\alpha),n}\} \to \alpha$.

$\square$

