# OpenReview forum: "On Testing Conditional Mean Independence for Manifold-Valued Data"
_ICML.cc/2026/Conference — ICML 2026 regular_

### Official Review · Reviewer_piYX · 2026-02-27

**Soundness:** 3
**Presentation:** 3
**Significance:** 3
**Originality:** 3
**Overall Recommendation:** 5
**Confidence:** 4

**Summary:**

In this paper the authors introduce a test for conditional means on Riemannian manifolds. To this end, the authors extend the classical null hypothesis to a hypothesis defined on the push forward density onto the tangent space at the mean using the log map. The authors then present asymptotic properties and a boostrap method for testing. The authors finish the paper with many simulations and an application.

**Compliance With Llm Reviewing Policy:**

Affirmed.

**Final Justification:**

My initial score of seems to accurately reflect the paper. Due to the discussion and reading the other reviews and rebuttals, my score seems to be reaffirmed.

**Key Questions For Authors:**

Is there a missing assumption on the range of the data? To have a unique mean, one would need to restrict the density of the data. Would the test statistic then be ill-defined if there are non-unique means?

What is the chosen metric for the SPD manifold? I did not see that detail, but I may have missed it.

Why was 499 chosen and not 500? This question is simply a curiosity.

**Limitations:**

Yes.

**Strengths And Weaknesses:**

The paper well written, the ideas are clear, the contribution is strong. Tying to this, a weakness is that the paper structure can be improved. The contributions are lumped together with previous literature. While a careful reader can disentangle the two, the authors could present it better.

The authors do many comparisons on a wide set of data generation methods, sufficient number of manifolds, and experimental data.

The authors are missing, in my opinion, a critical citation. Pennec's Intrinsic Statistics paper presents covariance as $E(\langle \log_\mu X,\log_\mu X \rangle)$, which is effectively definition 2.1 without the conditioning.

Figure 1 is not legible.

The manifold sections, while somewhat standard, should include more details on their geometric structure. For instance, I assume that for the sphere the canonical choice for the metric was used but it isn't explicitly stated.

Section 3 ends abruptly. Theorem 3.4 is important for experiments and should be discussed, even if briefly, after it is presented.

---

> ### Author Rebuttal · Authors · 2026-03-31
>
> Dear Reviewer,
>
> Thank you very much for your careful review and for the valuable comments and suggestions. We have read your feedback carefully and have addressed each point accordingly. Our detailed responses are provided below.
>
> I. Response to Weaknesses
> 1. Thank you for this suggestion. In the revised manuscript, we will reorganize the introduction by first reviewing the literature in a hierarchical manner: global independence tests → conditional mean independence tests → challenges with manifold-valued data. We will then add a transitional paragraph to clearly highlight the research gap, followed by a concise summary of our contributions. This restructuring will make the distinction between existing work and our contributions much clearer. We appreciate your valuable advice.
> 2. We sincerely thank you for pointing out the absence of this crucial reference. We fully agree that Pennec (2006)'s classic work on intrinsic statistics on Riemannian manifolds is an important theoretical foundation for statistical methods of manifold-valued data. In particular, the idea of defining covariance through tangent space linearization (i.e., the logarithmic map) is closely related to the core step of our method, which uses $\operatorname{Log}_\mu(Y)$ for tangent space projection. In the revised manuscript, we will add a Remark 2.3 in Section 2.2, specifically discussing the connection between our MMDD and Pennec's intrinsic covariance framework.
> 3. Thank you for your correction. Due to insufficient resolution and typesetting issues, some information in the original Figure 1 (presenting the covariate ranking of MMDD and dCov) is unclear. To improve the readability of data presentation, we will replace the results in Figure 1 with a table in the revised manuscript, detailing the statistic values, significance judgments, and rankings of each covariate for both MMDD and dCov.
> 4. Thank you for highlighting this issue. We will improve this as follows:
> (1) In Section 4 , we will briefly specify the metric for each manifold: $\mathbb{S}^2$ (standard induced Riemannian metric), Wasserstein manifold (2-Wasserstein metric), and $\operatorname{Sym}^+(2)$ (affine-invariant Riemannian metric, AIRM);
> (2) In the appendix, we wil provide complete geometric descriptions (manifold/tangent space definitions, metric expressions, logarithmic map formulas, and references) to ensure reproducibility.
> 5. Thank you for your pertinent comment. We will add a discussion after Theorem 3.4 in Section 3 to explain its key theoretical guarantee for the test’s consistency and the test power in the Wild Bootstrap implementation.
>
> II. Responses to Key Questions
> 1. Thank you for raising this important question. We acknowledge that the theoretical derivation of this paper is indeed based on the premise that the Fréchet mean exists and is unique. This assumption is a classic regularity convention in geometric manifold statistics and intrinsic mean inference, including Pennec (2006)'s classic work, whose theoretical framework also requires the data distribution to lie within a geodesically convex neighborhood to ensure the uniqueness of the Fréchet mean and the well-posedness of the logarithmic map. In applications, we assume the data distribution satisfies this condition (e.g., the data support is sufficiently concentrated or the manifold has non-positive sectional curvature). If the mean is not unique, the definition of $\operatorname{Log}_\mu(Y)$ depends on the choice of the mean, and the test statistic will no longer be well-defined, which constitutes a theoretical boundary and applicable limitation of our method. In the revised manuscript, we will explicitly add this assumption and discuss it as an applicable condition of the method.
> 2. The affine-invariant Riemannian metric (AIRM).
> 3. This is an interesting question. The choice of 499 bootstrap resamples is mainly for convenience in implementation. In practice, we combine the 499 bootstrap statistics with the original sample statistic, resulting in a total of 500 values, which facilitates stable empirical quantile calculation for critical values. This practice is common in the literature.

---

> > ### Author Rebuttal · Reviewer_piYX · 2026-03-31
> >
> > Thank you for the thorough response and explanation. I believe the authors have answered my questions and concerns sufficiently. My initial concerns and questions were not grave and hence my initial score reflected that standing.

---

### Official Review · Reviewer_teYA · 2026-03-07

**Soundness:** 3
**Presentation:** 3
**Significance:** 2
**Originality:** 3
**Overall Recommendation:** 5
**Confidence:** 3

**Summary:**

This manuscript aims to develop a new testing method for assessing conditional mean independence in manifold-valued data. The method is based on a measure called Manifold Martingale Difference Divergence (MMDD) and its corresponding empirical estimator.

**Compliance With Llm Reviewing Policy:**

Affirmed.

**Final Justification:**

The authors addressed my main concerns, so I have raised my score from 4 to 5

**Key Questions For Authors:**

1. In addition to the real data considered, could the author provide some other motivating real data examples to demonstrate the application scenarios of the methods?

2. Could the authors provide some sensitivity analysis results to the choice of kernels $K(X-X')$? A related question is how to choose the bandwidth in such models?

3. In the application, it appears that the authors applied their MMDD method to each individual covariate to rank their marginal impact. However, the MMDD is developed for any $X\in R^{P}$. Could the authors clarify on why only 1-d marginal test is used in the application?

4. Could the authors provide computational runtime comparisons (e.g., wall-clock time) for MMDD and the competing methods using different sample sizes?

**Limitations:**

Yes

**Strengths And Weaknesses:**

Strengths:
1. The paper considers an interesting extension of dependence testing to the context of manifold-valued data, where many existing methods are not directly applicable to.
2. The paper is in general clearly and well structured.
3. The proposed nonparametric methods have strong theoretical guarantees, including the asymptotic distribution derivations and the boostrap testing procedure.
4. The performance of the methods is compared against dCov-based tests, showing superior performance in various manifold spaces.

Weaknesses:
1. While the conditional mean independence test for manifold-valued data appears to be an interesting methodological problem, the topic itself appears to have a fairly narrow scope and relatively limited applications.
2. The proposed MMDD appears to be a relatively straightforward extension of the functional martingale difference divergence by replacing linear operations in Hilbert space to manifold data via tangent spaces.
3. The method relies on the choice of kernels $K(X,X')$. However, when $p$ is large, the choice of such kernels or kernel bandwidth becomes difficult. Moreover, the translation-invariant (e.g., stationary) kernel assumption may not hold in practice.
4. Although the theoretical computational complexity of MMDD appears to be faster than the dCov-based test, the algorithm itself appears to be still computationally expensive with  $O(n^2+Bn)$ complexity.

---

> ### Author Rebuttal · Authors · 2026-03-31
>
> Dear Reviewer,
>
> Thank you very much for your careful review and for the valuable comments and suggestions.
>
> I. Responses to Weaknesses
> 1. MMDD is not limited to conditional mean independence testing; it serves as a general tool in regression workflows:
> (1)Variable screening: ranking covariates via conditional mean dependence;
> (2)Model diagnostics: detecting residual dependence through tangent space projections, indicating model misspecification.
> 2. We acknowledge MMDD builds on FMDD’s core logic, but a natural extension does not equate to technical triviality—analogous to FMDD’s non-trivial extension of MDD (Shao & Zhang, 2014). Extending FMDD to manifolds poses fundamental technical challenges:
> (1) Nonlinear Fréchet mean: Manifolds lack global linear operations, requiring nonlinear optimization for Fréchet means; the $O_p(n^{-1/2})$ estimation error of $\hat{\mu}$ must be rigorously quantified.
> (2) Failure of U-centering: Linear U-centering $a_{ij} - a_i - a_j + a_-$ (for unbiased U-statistics in MDD/FMDD) is inapplicable to manifolds due to the absence of global subtraction and dependence of $\log_{\hat{\mu}}(Y_i)$ on $\hat{\mu}$.
> (3) Reconstructed asymptotic theory: We handle the estimation error of $\hat{\mu}$ via second-order expansion and prove the test statistic follows a non-central weighted chi-squared distribution (distinct from FMDD’s central form), a key technical contribution.
> 3. Kernel and bandwidth sensitivity is analyzed in detail in our response to Key Question 2. For high-dimensional covariates, MMDD retains test consistency but suffers from reduced power with increasing dimensions. To address this, we propose a 'marginal screening + joint testing'  two-step method: screening low-dimensional subsets of covariates with the strongest MMDD values, then conducting standard MMDD joint tests.
> 4. Although MMDD involves $O(n^2)$ pairwise computation, it is efficient in testing:
> (1)MMDD: precompute matrix once; bootstrap requires only $O(n)$  quadratic forms;
> (2)dCov: recomputes distances per permutation ($O(n^2)$ each time).
>
> II. Responses to Key Questions
> 1. We will expand discussion to include:Medical imaging: Conditional mean relationships between SPD manifold-valued DTI data and clinical variables;  Shape analysis: Associations between human/organ shapes (Kendall shape space) and demographic variables; Robotics: Dependence between end-effector poses ($SO(3)/SE(3)$) and control signals.
> 2. We conducted sensitivity analysis (p=5, n=100) with Gaussian/Laplacian kernels.
> Main findings: Kernel choice is robust, performance trends are consistent; Bandwidth is critical,  small → size inflation, large → conservative/degenerate; Median heuristic fails (bandwidth ≈ 2.8–3.1) due to mismatch with dependence scale.
> We adopt a data-driven selection strategy:Silverman rule for initial scaling;size-calibrated grid search; optionally, power maximization + permutation calibration.
>
> Table 1 $r^2=0,s^2=0.2$ size (rows 1–5) and $r^2=0.5,s^2=0$ power
> | Bandwidth    | Gaussian rej0.1​/rej0.05​/rej0.01​ | Laplacian rej0.1​/rej0.05​/rej0.01​ |
> | ----------------- | ----------------------------------------------- | -------------------------------------------------- |
> | 0.10              | 0.204/0.055/0.008                               | 0.138/0.032/0.006                                  |
> | 0.25              | 0.150/0.044/0.007                               | 0.100/0.039/0.005                                  |
> | 0.50              | 0.096/0.036/0.004                               | 0.038/0.011/0.002                                  |
> | 0.75              | 0.057/0.025/0.000                               | 0.016/0.006/0.000                                  |
> | median (2.911/2.993) | 0.000/0.000/0.000                               | 0.000/0.000/0.000                                  |
> | 0.10              | 0.291/0.017/0.005                               | 0.531/0.243/0.011                                  |
> | 0.25              | 0.484/0.204/0.008                               | 0.942/0.886/0.606                                  |
> | 0.50              | 0.948/0.894/0.624                               | 1.000/0.997/0.982                                  |
> | median (2.795/3.078) | 0.982/0.891/0.212                               | 0.986/0.864/0.175                                  |
>
> 3. Marginal MMDD is used for variable ranking and interpretability. MMDD fully supports joint testing (already used in residual tests).
> Following your suggestion, we will revise the workflow: first conduct joint testing; then apply marginal analysis to identify key variables.
> 4. We will include runtime comparisons. Preliminary results for DGP1 on the $\mathbb{S}^2$ sphere ($s^2=0.5$):
> | Sample size (n) | MMDD runtime | dCov runtime |
> | --------------- | ------------- | ------------ |
> | 50              | 0.086         | 2.356        |
> | 100             | 0.089         | 2.247        |
> | 200             | 0.092         | 2.259        |

---

> > ### Author Rebuttal · Reviewer_teYA · 2026-03-31
> >
> > The authors have addressed my concerns, and I will raise the score

---

### Official Review · Reviewer_Bc8W · 2026-03-11

**Soundness:** 4
**Presentation:** 3
**Significance:** 2
**Originality:** 2
**Overall Recommendation:** 4
**Confidence:** 3

**Summary:**

The paper proposes a nonparametric test for conditional mean independence between a manifold-valued response and Euclidean predictors via the MMDD, which generalizes to manifold-valued data by projecting onto tangent spaces, combining tangent-space inner products with kernel. Asymptotic properties under the null and local alternatives are studied, and a mutiplier bootstrap approach is introduced for computation. The methods are illustrated through simulations, as well as an application to wind direction data.

**Compliance With Llm Reviewing Policy:**

Affirmed.

**Final Justification:**

The authors have sufficiently explained the technical challenges in the work during the rebuttal, which has increased my confidence in the paper. Therefore, I revised my score accordingly.

**Key Questions For Authors:**

1. Can the authors clarify how the proposed MMDD could be integrated into learning pipelines, e.g., as a diagnostic for geometric neural networks or generative models on manifolds?

2. Are there scalable approximations or randomized variants that would make the method practical for high-dimensional or large-sample datasets?

3. Could MMDD be reinterpreted through kernel or energy distance embeddings to connect with existing ML tools such as kernel independence tests or HSIC?

**Limitations:**

No.

**Strengths And Weaknesses:**

**Strengths:**

1.  The theoretical development is mathematically solid, clearly grounded in the Fréchet mean and tangent-space tools from Riemannian statistics. Proofs and asymptotic arguments are detailed and self-consistent.

2. Extending MDD to manifold-valued responses is nontrivial.

3. Comprehensive simulations on multiple manifolds demonstrate the proposed test’s size and power performance.

**Weaknesses:**

1. The work is rooted in mathematical statistics, not machine learning, focusing on an asymptotic testing procedure rather than a learning algorithm or optimization method. Its motivation and applications emphasize statistical inference, with limited relevance to ML concerns like generalization, scalability, or algorithms.

2. MMDD is not directly connected to ML frameworks such as manifold learning, geometric deep learning, or causal representation learning, making it an elegant but isolated statistical contribution.

3. The writing is dense and highly technical, with long asymptotic derivations that obscure intuition. Clearer separation of conceptual insight from proofs would help ML readers understand why MMDD matters and when it is useful.

4. MMDD extends functional MDD via standard Riemannian adaptations, offering incremental methodological contribution rather than a paradigm shift for ML practice.

---

> ### Author Rebuttal · Authors · 2026-03-31
>
> Dear Reviewer,
>
> Thank you very much for your careful review and for the valuable comments and suggestions.
>
> I. Responses to Weaknesses
>
> 1. As geometric data (spheres, SPD matrices, probability distributions) grow prevalent in ML—e.g., spherical data in meteorology/robotics, SPD matrices in neuroscience/medical imaging, and distributions in generative models—there is an urgent need for statistical diagnostic tools tailored to such data. Current ML lacks diagnostics for manifold-valued responses + Euclidean covariates tasks, and MMDD fills this gap as an integrable ML workflow tool:
> (1) Geometric NN diagnostics: Post-training, compute MMDD between residuals (log-mapped differences of predictions vs. ground truth) and covariates. A significant MMDD indicates unlearned conditional mean structures, guiding architecture refinement or regularization.
> (2) Manifold generative model evaluation: Beyond global metrics (FID, MMD), MMDD assesses if generated samples’ conditional Fréchet means align with true counterparts, enabling fine-grained evaluation.
> (3) Variable screening for model simplification: Preprocess high-dimensional covariates to identify impactful ones, reducing input dimensionality and improving training efficiency.
> 2. As detailed in our responses to Weaknesses 1.
> 3. Thank you for this reminder. We fully agree that core ideas and practical value should be prioritized over lengthy derivations. Our current structure already addresses this: Full proofs of all theorems are relegated to the appendix. The main text retains only key conclusions and actionable method descriptions. To further improve accessibility, we will add a "Method Overview" at the start of the main text in the revised manuscript, using plain language to explain MMDD’s core intuition and application scenarios for ML-background readers.
> 4. We acknowledge MMDD builds on FMDD’s core logic, but a natural extension does not equate to technical triviality—analogous to FMDD’s non-trivial extension of MDD (Shao & Zhang, 2014). Extending FMDD to manifolds poses fundamental technical challenges:
> (1) Nonlinear Fréchet mean: Manifolds lack global linear operations, requiring nonlinear optimization for Fréchet means; the $O_p(n^{-1/2})$ estimation error of $\hat{\mu}$ must be rigorously quantified.
> (2) Failure of U-centering: Linear U-centering $a_{ij} - a_i - a_j + a_-$ (for unbiased U-statistics in MDD/FMDD) is inapplicable to manifolds due to the absence of global subtraction and dependence of $\log_{\hat{\mu}}(Y_i)$ on $\hat{\mu}$.
> (3) Reconstructed asymptotic theory: We handle the estimation error of $\hat{\mu}$ via second-order expansion and prove the test statistic follows a non-central weighted chi-squared distribution (distinct from FMDD’s central form), a key technical contribution.
>
> II. Responses to Key Questions
> 1. As detailed in our responses to Weaknesses 1.
> 2. This is a critical question. MMDD’s current $O(n^2)$ complexity poses challenges for large samples. We propose two scalable extensions:
> Approximation strategies: For large $n>500$, subsampling or random projection reduces complexity to $O(n\sqrt{n})$ or $O(nk)$ (k = projection dimension).
> Random Fourier Features (RFF): For translation-invariant kernels (e.g., Gaussian), RFF approximates the kernel as a finite-dimensional inner product, reducing complexity to $O(nd)$ (d = number of RFF).
> 3. This is an insightful question. MMDD has close theoretical links to existing ML tools:
> Kernel embedding (vs. HSIC): HSIC tests global independence via RKHS embeddings of both $X$ and $Y$. MMDD tests conditional mean independence by embedding manifold responses into tangent spaces (local Euclidean spaces) via $\langle \log_{\hat{\mu}}(Y_i), \log_{\hat{\mu}}(Y_j) \rangle$, capturing mean-based dependence (not global distributional dependence) for manifold data.
> Energy distance (vs. MDD): MMDD generalizes MDD to manifold-valued responses. For energy distance kernels ($K(X_i,X_j)=\|X_i-X_j\|$), MMDD uses the same covariate similarity metric as MDD. Instead of linear space-centered terms $\{V-\mathbb{E}(V)\}$, MMDD adopts tangent space inner products to characterize mean differences of manifold responses.

---

> > ### Author Rebuttal · Reviewer_Bc8W · 2026-04-01
> >
> > While I still find the connection of the theoretical contribution of this paper to any practical applications somewhat limited, I appreciate the authors’ technical rigor and their time and efforts spent in explaining the challenges in the proof. In light of the authors’ willingness to reorganize the paper to improve its accessibility and readability for a broader machine learning audience, I am inclined to raise my recommendation to Weak Accept.

---

### Official Review · Reviewer_MLKN · 2026-03-12

**Soundness:** 3
**Presentation:** 4
**Significance:** 2
**Originality:** 1
**Overall Recommendation:** 3
**Confidence:** 4

**Summary:**

The paper extends the Martingale Difference Divergence to manifold-valued responses (MMDD), making use of the logarithmic projection of the response onto the tangent space at Frèchet mean. The authors establish the asymptotic null distribution, relying on an infinite sum of weighted chi-squared variables based on U-statistic theory, demonstrate local power, and propose a consistent bootstrap procedure for finite-sample inference. Simulations on the sphere, Wasserstein space, and SPD manifolds validate the method, as well as a real data application for variable selection and fit evaluation in the context of Frèchet regression.

**Compliance With Llm Reviewing Policy:**

Affirmed.

**Key Questions For Authors:**

1. A light, minor comment: why use $\mu$ for the unconditional FM, and $m_X$ for the conditional one?

2. In Assumption 2.3, you should specify that such expansion is not always valid; in particular, $A$ could be non-negative, and not necessarily positive-definite. In such cases, one can still hope to get a limiting distribution, but things get more tricky, and this should be mentioned.

3. The consistency established is restricted to the family of alternatives defined in Eq. (6). While this class is broad, it is not fully general, and decomposes dependence on $X$ and the noise in a specific form. The manuscript would be strengthened by a brief discussion on whether the test remains consistent under a more general non-parametric alternative.

4. Since your test relies on a kernel function, could you explore its behavior in higher-dimensional settings? Kernel-based tests are known to sometimes bypass the curse of dimensionality in their estimation rates. Investigating whether MMDD inherits this property (either theoretically or via a simulation varying $p$) would add significantly to the value of the paper, in my opinion.

5. To strengthen the theoretical contribution, could the authors better highlight any specific mathematical challenges that arose specifically from the manifold geometry beyond standard U-statistic theory?

6. It would be helpful if the authors could clarify whether the absence of curvature terms is a genuine theoretical property of the MMDD construction, or whether it is an artifact of the tangent-space approximation around the global Fréchet mean, and the assumption of validity of the taylor expansion for the clt.

**Limitations:**

Yes.

**Strengths And Weaknesses:**

1. The paper is well written and accessible. It  targets a relevant problem in manifold statistic, and proposes an intrinsic extension of a well-known method to test conditional dependence. I particularly appreciated that the authors are able to show non-trivial power even for an adversarial family of alternatives with mean-level dependance.

2. The simulations are comprehensive, evaluating the method across three diverse and representative manifolds and shows good power against both linear and nonlinear alternatives
The application to Fréchet regression residuals  is a very practically useful contribution.
The implementation of the wild bootstrap is light.

3. The theoretical framework is rigorous and sound, though appears to follow from on existing literature, in particular the proofs for the null distribution and bootstrap consistency rely on classic U-statistic expansions. In particular, though the problem is posed on manifolds, the method essentially projects responses to the tangent space at the global Fréchet mean and then applies a Euclidean dependence test. After this projection, the statistic and its asymptotic analysis rely entirely on standard inner products and classical U-statistic theory, so the manifold structure plays only a minimal role.

4. The paper points out that Distance Covariance (dCov) fails when the variance (second-order) of the response depends on the covariate, even if the mean does not. While the simulations show the proposed MMDD test is robust to this aspect, the consistency results  only cover homoschedastic alternatives.

---

> ### Author Rebuttal · Authors · 2026-03-31
>
> Dear Reviewer,
>
> Thank you very much for your careful review and for the valuable comments and suggestions.
>
> Responses to Key Questions
> 1. We follow standard practice in Fréchet regression (e.g., Petersen & Müller, 2019) but use $\mu$ for the unconditional Fréchet mean to enhance interpretability, as $\mu$ commonly denotes a global mean. The conditional mean is denoted by $m_X$ to avoid confusion with $\mathbb{E}(X)$, which is often written as $\mu_X$.
> 2. We agree this is an important point. The definiteness of $A$ is closely related to the uniqueness of the Fréchet mean, which holds when the distribution lies in a geodesically convex neighborhood (Pennec, 2006; Bhattacharya & Patrangenaru, 2003). Under this condition, $A$ is typically positive definite and $\hat{\mu}$ achieves $\sqrt{n}$-consistency. If this condition fails, $A$ may be only semi-definite, the Fréchet mean may be non-unique, and convergence rates as well as asymptotic distributions become more complex. We will clarify this assumption and its limitations in the revision.
> 3. Our theoretical result (Theorem 3.4) is established under local alternatives with independent noise, which is standard for analyzing local power. However, our simulations consider more general models:  $\log_\mu(Y) = g(X) + h(X)\varepsilon$,
>  allowing heteroscedasticity and dependence between signal and noise. Across multiple manifolds (sphere, Wasserstein, SPD), MMDD shows strong empirical power. Thus, while the theory focuses on local alternatives, simulations demonstrate robustness under broader nonparametric settings.
> 4. MMDD shares structural similarities with kernel-based tests (e.g., HSIC). In high dimensions:
> (1) Consistency is retained, but required sample size increases;
> (2) Power decreases due to distance concentration, causing kernel degeneracy;
> (3) Kernel methods do not eliminate the curse of dimensionality, but can only mitigate it;
> (4) Recent studies (Yan & Zhang, 2021; Zhang, 2021; Zhu et al., 2020) have demonstrated that kernel-based test statistics can only characterize linear dependence between Y and X in high-dimensional scenarios. Accordingly, even with appropriately chosen kernels, MMDD is also restricted to detecting linear conditional mean dependence in high-dimensional scenarios.
>  MMDD as a variable screening tool: 'screening + testing' two-step method:
> To address high-dimensional covariates, we propose an efficient solution—using MMDD’s screening capability for dimensionality reduction, followed by joint testing on the reduced subset:
> (1) Screening (MMDD for variable selection)
>   Compute the standardized MMDD value (scaled to [0,1]) between each covariate $X_j$ ($j=1,\dots,p$) and response $Y$, rank by MMDD values, and select the top-k covariates with the strongest conditional mean dependence (reducing dimension from p to k).
> (2) Joint testing (MMDD for model testing)
>   Conduct standard MMDD joint testing on the selected low-dimensional subset to verify conditional mean independence. The reduced dimension avoids distance concentration, allowing direct use of Gaussian kernels with median heuristic or small bandwidth.
>
> 5. Thank you for this critical question. We acknowledge the original manuscript can better emphasize technical challenges from manifold geometry. In fact, handling manifold data poses the following non-trivial challenges:
> (1) Fundamentally changed mean definition: FMDD relies on linear global expectations (Hilbert space); manifolds lack global addition/subtraction, requiring nonlinear Fréchet means (solved via optimization). The Fréchet mean is not a simple sample average, and its empirical estimate $\hat{\mu}$ introduces a non-negligible $O_p(n^{-1/2})$ error that must be rigorously quantified.
> (2) U-centering failure: MDD and FMDD use U-centering (e.g., $a_{ij} - a_i - a_j + a_-$) to construct unbiased U-statistics, eliminating first-order terms for a central weighted chi-squared asymptotic distribution. This method cannot be directly extended to manifolds: no global subtraction exists, and residuals $\log_{\hat{\mu}}(Y_i)$ depend on $\hat{\mu}$, making linear U-centering ineffective.
> (3) Reconstructed asymptotic theory: Nonlinear manifold structures and $\hat{\mu}$’s estimation error prevent direct reuse of FMDD’s asymptotics. In Theorem 1’s proof, we explicitly handle $\hat{\mu}$’s error via second-order expansion and prove the test statistic follows a non-central weighted chi-squared distribution (distinct from FMDD’s central form)—our core technical contribution.
> 6. Curvature does not disappear but is implicitly incorporated via the Hessian operator in the second-order expansion. Specifically, curvature enters the asymptotic distribution through terms involving $\mathbb{E}(\mathcal{H}_{i,j})$, affecting the eigenvalues of the limiting distribution.

---

> > ### Author Rebuttal · Reviewer_MLKN · 2026-04-03
> >
> > I acknowledge that many of my previous concerns and remarks have been addressed in the rebuttal. However, it still appears that the “manifold-specific” aspects primarily enter through a Taylor expansion around the tangent space, which reduces the asymptotic analysis to a classical Euclidean CLT and is assumed upstream. I remain doubtful that the manifold geometry substantively affects the test statistic beyond this tangent-space linearization. Could the authors clarify or point to specific technical aspects I may have missed that pertain to the manifold structure, beyond the general (and well-understood) challenges arising from the lack of linearity?

---

> > > ### Author Response · Authors · 2026-04-04
> > >
> > > Dear Reviewer,
> > >
> > > Thank you for your continued insightful questions. We understand your concern: Does the manifold geometry genuinely affect the construction, estimation, and behavior of MMDD beyond tangent-space linearization? Below we explain from three aspects.
> > > 1. Definition: Two core components of MMDD are fully determined by manifold geometry
> > >
> > > (1) Fréchet mean $\mu = \arg\min_{\omega \in \mathcal{M}} \mathbb{E}[d_\mathcal{M}^2(Y,\omega)]$: This is not a linear expectation but a geometric center defined by the manifold metric. Different manifolds (sphere, SPD, Wasserstein) have different metrics, and thus the computation of the mean differs. Its existence and uniqueness require manifold‑specific regularity conditions (geodesic convexity, bounded curvature).
> > > (2) Logarithmic map $\log_\mu(Y)$: It is not merely a device to handle the lack of linear operations; it deeply embeds the manifold’s core geometric structure. Its definition strictly depends on geodesics and the metric, and its explicit form varies across manifolds.
> > >
> > > 2. Estimation: Manifold geometry alters estimation methods and statistical properties
> > >
> > > (1) Empirical Fréchet mean $\hat{\mu}$: To compute $\hat{\mu} = \arg\min_{\omega \in \mathcal{M}} \frac{1}{n}\sum_{i=1}^n d_\mathcal{M}^2(\omega, Y_i)$, the estimation process fully relies on manifold geometry:
> > > - It has no closed-form solution and adopts Riemannian gradient descent. The iterative process is implemented through 'gradient computation in the tangent space + exponential mapping to pull back to the manifold' — the exponential mapping is the inverse operation of the logarithmic map, whose definition relies on the geodesics of the manifold (Pennec, 2006), and the gradient descent update rules vary completely across different manifolds.
> > > - Convergence guarantee: The $\sqrt{n}$-consistency of $\hat{\mu}$ requires the positive definiteness of the operator $A = \mathbb{E}[\nabla_\mu^2 d(\mu,Y)^2]$ (Bhattacharya & Patrangenaru, 2003; 2005). The positive definiteness of $A$ is directly determined by curvature and geodesic convexity. If curvature is too large or the data lie outside a geodesically convex neighborhood, $A$ may be only positive semi‑definite, slowing convergence below $\sqrt{n}$ and affecting finite‑sample performance. In Assumption 2.3, we represent the estimation error of $\hat{\mu}$ (which is of order $O_p(n^{-1/2})$) in the Bahadur form $\log_\mu \hat{\mu} = A^{-1} \frac{1}{n}\sum_i \log_\mu Y_i + o_p(n^{-1/2})$, thereby explicitly propagating this error into the MMDD statistic. This step is a technical difficulty.
> > >
> > > In contrast, the sample mean in MDD/FMDD is simply the linear average $\bar{Y} = \frac{1}{n}\sum Y_i$, requiring no optimization and no geometric constraints.
> > >
> > > (2) Failure of U‑centering: Manifolds lack global subtraction, so we cannot construct unbiased U‑statistics via linear centering as in MDD/FMDD. We are forced to directly handle the estimation error of $\hat{\mu}$, which leads to a non‑central asymptotic distribution.
> > >
> > > 3. Applicability: Curvature directly restricts the valid domain of MMDD
> > > According to Pennec (2006), the uniqueness of the Fréchet mean and the well‑definedness of the log map require that the data support lie in a geodesically convex neighborhood of radius $r < \pi/(2\sqrt{\kappa})$ (when $\kappa>0$). For example, on the unit sphere ($\kappa=1$), all data must lie within some open hemisphere. This is a substantive restriction imposed by curvature; if violated, MMDD may fail.
> > >
> > > Summary
> > > Manifold geometry does not 'disappear' after tangent‑space linearization; rather:
> > > - Definitionally, geometric properties are deeply embedded through the Fréchet mean and the logarithmic map;
> > > - Estimation‑wise, it forces nonlinear optimization and induces a non‑central asymptotic distribution;
> > > - Applicability, it imposes curvature‑dependent constraints on the valid domain;
> > > - Power, it indirectly affects testing performance through the stability of these core components (even if not explicitly separable).
> > >
> > > Thank you again for your deep and constructive question.

---

### Decision · Program_Chairs · 2026-04-30

**Decision:**

Accept (regular)

**Comment:**

This paper introduces MMDD, a nonparametric test for conditional mean independence between manifold-valued responses. MMDD will have potential impact on relevant ML workflows. Most of the reviewers note that the theoretical development is mathematically solid and the paper provides novel methodology in MMDD. Nearly all of the reviewers' concerns were addressed during the rebuttal period. One reviewer noted that the role of the manifold geometry seems largely absorbed by some linearization techniques. However, during the reviewer discussion period, another reviewer pointed out that since manifolds are defined as being locally linear, these linearizations do make sense. During this discussion, it was also mentioned that a longer discussion about the usage of curvature in the method would strengthen the paper.